

# Mixing and air-sea buoyancy fluxes set the time-mean overturning circulation in the subpolar North Atlantic

Dafydd Gwyn Evans[1], N. Penny Holliday[1], Sheldon Bacon[1], and Isabela Le Bras[2]

[1]National Oceanography Centre, Southampton, UK
[2]Woods Hole Oceanographic Institute, Woods Hole, Massachusetts, USA

**Correspondence:** Dafydd Gwyn Evans (dgwynevans@noc.ac.uk)

**Abstract.** The overturning streamfunction as measured at the OSNAP (Overturning in the Subpolar North Atlantic Program) mooring array represents the transformation of warm/salty Atlantic Water into cold/fresh North Atlantic Deep Water (NADW). The magnitude of the overturning at the OSNAP mooring array can therefore be linked to the water mass transformation by air–sea buoyancy fluxes and mixing in the region to the north of the OSNAP array. Here, we estimate these water mass trans-

formations using a combination of observational-based, reanalysis-based and model-based datasets. Our results highlight the complementary roles of air–sea buoyancy fluxes and mixing in setting the time-mean magnitude of the overturning at OSNAP. A cooling by air–sea heat fluxes and a mixing-driven freshening in the Nordics Seas, Iceland Basin and Irminger Sea, precondition the warm/salty Atlantic Water, forming subpolar mode water classes. Mixing in the interior of the Nordic Seas, over the Greenland-Scotland ridge and along the boundaries of the Irminger Sea and Iceland Basin drive a water mass transformation

that leads to the convergence of volume in the water mass classes associated with NADW. Air–sea buoyancy fluxes and mixing therefore play key and complementary roles in setting the magnitude of the overturning within the subpolar North Atlantic and Nordic Seas. This study highlights that for climate models to realistically simulate the overturning circulation in the North Atlantic, the small scale processes that lead to the mixing-driven formation of NADW must be adequately represented within the model's parameterisation scheme.

## 1  Introduction

The subpolar North Atlantic and the Nordic Seas are key regions for the formation of the water masses that ventilate the deep ocean (Daniault et al., 2016; Johnson et al., 2019). As part of the global meridional overturning circulation (MOC), these deep water masses regulate the global budgets of heat and carbon by sequestering excess heat and carbon dioxide into the deep ocean (Pérez et al., 2010; Purkey and Johnson, 2010; Khatiwala et al., 2013; Fröb et al., 2016). The North Atlantic Deep Water

(NADW) that propagates out of the subpolar North Atlantic is composed of a combination of dense overflow waters that form in the Nordic Seas, and the slightly less dense water masses formed in the Irminger and Labrador Seas (Bower et al., 2019). The strength of the MOC as measured in the subpolar North Atlantic is directly related to the diapycnal water mass transformations that form the dense waters which make up NADW.



The North Atlantic Current (NAC) brings warm, salty and light Atlantic Water into the eastern branch of the North Atlantic
subpolar gyre. The pathways of the NAC determine the magnitude of the water mass transformation that ultimately forms
NADW (Daniault et al., 2016; Bower et al., 2019). The Atlantic Water that circulates within the subpolar gyre eventually forms
Labrador Sea water (LSW) and the water masses formed via convection in the Labrador and Irminger Sea. These water masses
classically compose the lighter classes of upper NADW. On the other hand, the Atlantic Water that continues northward into
the Nordic Seas typically forms overflow waters that combine to form the denser lower NADW.

Along the pathways of the subpolar gyre, Atlantic water experiences seasonal cooling to form subpolar mode water (Petit
et al., 2020). Further cooling in the Irminger and Labrador Seas leads to open ocean convection and the subsequent formation of
LSW. In order to affect the MOC strength and ventilate the ocean interior as upper NADW, this LSW must be incorporated into
the boundary currents of the Irminger and Labrador Seas (Brüggemann and Katsman, 2019; Le Bras et al., 2020; MacGilchrist
et al., 2021). This exchange between the boundary currents and interiors of the subpolar basins drives an additional mixing-
driven water mass transformation key to the formation of NADW.

Alternatively, the warm, salty and light Atlantic water continues northward into the Nordic Seas. Here it is cooled and
freshened as it circulates through the Nordic Seas, forming intermediate and dense water mass classes (Mauritzen, 1996;
Tsubouchi et al., 2021). These water masses ultimately provide the source for the overflow waters that enter the subpolar basin,
but are generally confined to the Nordic Seas by the relatively shallow Greenland-Scotland ridge (Mauritzen, 1996; Voet et al.,
2010; de Jong et al., 2018; Bower et al., 2019). Upon crossing the Denmark Strait, Iceland-Scotland Ridge and Faroe-Bank
Channel to descend into the subpolar basin, these overflow waters mix with overlying less dense water masses, modifying their
temperature and salinity. Entrainment of warmer and saltier subpolar mode water in the Iceland Basin, ultimately distinguishes
Iceland-Scotland Overflow Water (ISOW) from Denmark Strait Overflow Water (DSOW). DSOW and ISOW subsequently
flow cyclonically along the deep boundary currents of the subpolar North Atlantic and ultimately make up the densest lower
NADW that is exported southward from the subpolar North Atlantic.

The net effect of the diapycnal water mass transformation, or the diapycnal overturning, that forms LSW, DSOW and ISOW
in the subpolar North Atlantic is estimated indirectly at the OSNAP (Overturing in the Subpolar North Atlantic Program)
mooring array (Lozier et al., 2017). At OSNAP, a diapycnal overturning streamfunction is approximated by accounting for the
northward flow of light water, and the southward flow of dense water masses. The maximum value of this diapycnal overturning
streamfunction represents the magnitude and variability of the diapycnal water mass transformation that forms dense water in
the subpolar North Atlantic and the Nordic Seas. The transport across the OSNAP mooring array indicates that the diapycnal
water mass transformation in the eastern subpolar basin (i.e. north of the OSNAP east mooring array from Greenland to
Scotland) sets the magnitude and variability of the overturning across the whole subpolar basin (including the OSNAP west
mooring array between the Newfoundland and Greenland; Lozier et al., 2019). A density compensating freshening counters
the strong cooling of water in the Labrador Sea, resulting in limited diapycnal transformation and therefore little diapycnal
overturning across the OSNAP west mooring array (Zou et al., 2020).

Recent work has predominantly focused on the role of air–sea buoyancy fluxes in driving the water mass transformation
at the density of maximum overturning as measured at the OSNAP mooring array (Desbruyères et al., 2019; Petit et al.,





2020). Petit et al. (2020) highlight that the water mass transformation by air–sea buoyancy fluxes at the density of maximum overturning in the Iceland Basin and Irminger Sea is similar in magnitude to the net water mass transformation implied by the overturning streamfunction measured at the OSNAP mooring array. Further, Desbruyères et al. (2019) show that the annual mean overturning at 45°N correlates with the 5 year lagged variability of the annual mean surface forced overturning (i.e. the maximum of the diapycnal water mass transformation by air–sea fluxes) north of 45°N. However, as part of their analysis, Desbruyères et al. (2019) and Petit et al. (2020) do not consider their estimates of the diapycnal transformation by air–sea fluxes in terms of the total volume change in density classes within their domain (Walin, 1982). They therefore do not account for the role of mixing in driving a diapycnal overturning in the subpolar North Atlantic, conflicting with our established understanding of the role of mixing in setting the properties of NADW via boundary current exchange and overflow water entrainment (Brüggemann and Katsman, 2019; Le Bras et al., 2020; Xu et al., 2018). Further, existing direct or inferred estimates of mixing rates in the subpolar North Atlantic are region specific or sparse in space and time (e.g. Lauderdale et al., 2008; Fer et al., 2010; Beaird et al., 2012), limiting our ability to quantify the role of mixing in setting the strength of the subpolar overturning.

To fully understand the drivers of AMOC strength and variability, it is therefore critical to characterise and quantify the roles of both the air–sea buoyancy fluxes and the mixing in setting the diapycnal overturning in the subpolar North Atlantic and Nordic Seas. Without a clear understanding of how air–sea buoyancy fluxes and mixing combine to form NADW, we cannot accurately simulate these processes in ocean-climate models and therefore understand their sensitivity to a changing climate. To address the combined roles of air–sea fluxes and mixing in setting the overturning streamfunction as measured at OSNAP, here we use a comprehensive water mass transformation framework to describe the diabatic processes that set the magnitude of the diabatic overturning in the subpolar North Atlantic. Building on the work of Walin (1982) and Speer and Tziperman (1992), and more recently Evans et al. (2014, 2017, 2018) and using a combination of observational- and model-based data, we will estimate the diabatic transformation by air-sea buoyancy fluxes and mixing, and compare to the strength of the overturning streamfunction as measured at the OSNAP mooring array.

## 2    Data and Methods

### 2.1    The water mass transformation framework

Walin (1982) introduced a framework to diagnose the strength of the overturning circulation from air-sea buoyancy fluxes within in a fixed geographical domain. They showed that the rate of change of volume between two tracer surfaces (i.e. $C^* \pm \Delta C/2$, where $C$ is a generic tracer that could be temperature ($\Theta$), salinity ($S$) or potential density ($\rho$), and $^*$ indicates the iso-surface of that tracer) is linked to the transport ($M$) into the domain, and the divergence of the water mass transformation between two tracer surfaces ($G$):

$$\frac{\mathrm{d}V(C^*,t)}{\mathrm{d}t} = M + \frac{\partial G(C^*,t)}{\partial C}.$$

(1)





The volume ($V$) is the amount of water in a tracer bin defined by $C^* \pm \Delta C/2$ and within a fixed geographical domain is given by

$$V(C^*, t) = \iiint \Pi[C, C^*] \, \mathrm{d}x \mathrm{d}y \mathrm{d}z, \tag{2}$$

where $\Pi[C, C^*]$ is a boxcar function that is equal to 1 if $C = C^* \pm \Delta C/2$, and is otherwise 0. If part of the fixed geographical domain is terminated at some point by a fixed section, then the volume rate of change within the tracer bin $C^* \pm \Delta C/2$ is given by

$$M(C^*, t) = \iint \Pi[C, C^*] \, v \, \mathrm{d}x \mathrm{d}z, \tag{3}$$

where $v$ is the velocity normal to the fixed section.

The water mass transformation into a given tracer bin can be represented as

$$G(C^*, t) = g(C^* - \Delta C/2, t) - g(C^* + \Delta C/2, t), \tag{4}$$

where $g$ represents the transformation across a given tracer surface:

$$g(C^* - \Delta C/2, t) = \int_{C^* - \Delta C/2} \frac{1}{|\nabla C|} \frac{\partial C}{\partial t} + \mathbf{u} \cdot \frac{\nabla C}{|\nabla C|} \, \mathrm{d}A. \tag{5}$$

The terms on the right hand side of Eq. (5) are integrated over the surface area of a tracer surface where $C = C^* - \Delta C/2$. Further, $\mathbf{u}(x, y, z, t)$ represents the three-dimensional velocity field. Within this framework, the only processes that can affect the transformation of water across a tracer surface, and consequently the volume within $C = C^* \pm \Delta C/2$, are water mass transformations due to air-sea buoyancy fluxes ($E$), where that tracer surface outcrops at the sea surface, and mixing ($F$), so that

$$G(C^*, t) = E(C^*, t) + F(C^*, t), \tag{6}$$

$F$ is derived as a residual of the known transformation $G$ and the air–sea flux driven transformation $E$. As in Eq. (4), the terms on the right hand side of Eq. (6) represent the difference between the transformation across the tracer bin $C^* \pm \Delta C/2$, for example:

$$E(C^*, t) = e(C^* - \Delta C/2, t) - e(C^* + \Delta C/2, t). \tag{7}$$

To define the contribution for air-sea fluxes, we must diverge from a general representation, and focus on specific tracers. The diapycnal transformation by air-sea buoyancy fluxes across an isopycnal surface is given by

$$e(\rho^*, t) = \frac{1}{\Delta \rho} \int \Pi[\rho, (\rho^* - \Delta \rho/2) \pm \Delta \rho/2] \left[ -\frac{\alpha}{C_p} q_{net} - \beta \frac{S}{1 - S} f_{net} \right] \mathrm{d}A, \tag{8}$$

where $\alpha$ is the thermal expansion coefficient, $\beta$ is the haline contraction coefficient, $C_p$ represents the specific heat capacity of seawater, $q_{net}$ is the net surface heat flux and $f_{net}$ is net surface freshwater flux. Here the right hand side of Eq. (8) is integrated over the surface area of the ocean where the density bin $\rho^* \pm \Delta \rho/2$ outcrops.



Speer and Tziperman (1992) expanded the framework of Walin (1982) to incorporate the diathermal transformation by air–sea heat fluxes:

$$e(\Theta^*, t) = \frac{1}{\rho C_p \Delta \Theta} \int \Pi[\Theta, (\Theta^* - \Delta\Theta/2) \pm \Delta\Theta/2] \, q_{net} \, \mathrm{d}A, \qquad (9)$$

and the diahaline transformation by air–sea freshwater fluxes:

$$e(S^*, t) = \frac{1}{\Delta S} \int \Pi[S, (S^* - \Delta S/2) \pm \Delta S/2] \, f_{net} S \, \mathrm{d}A. \qquad (10)$$

Evans et al. (2014) expanded the above frameworks into two-dimensional tracer space (i.e. $(\Theta, S)$ space) so that Eq. (1) becomes

$$\frac{\mathrm{d}V(\Theta^*, S^*, t)}{\mathrm{d}t} = M(\Theta^*, S^*, t) + \frac{\partial G(\Theta^*, t)}{\partial \Theta} + \frac{\partial G(S^*, t)}{\partial S}. \qquad (11)$$

As described in Evans et al. (2014), Eqs. (2), (3) and (8)-(10) can then be expanded into two-dimensional tracer space by including a boxcar function for each tracer.

$V$ is calculated within the desired domain using a gridded dataset of $\Theta$ and $S$, while $M$ is calculated using a section of $\Theta$, $S$ and $v$ from the open boundaries of the domain. From $M$ and the change in $V$ the transformation $G$ can be derived (Evans et al., 2014). Given the air–sea heat and freshwater fluxes, the air–sea flux driven transformation $E$ is calculated, and following

Eq. (4), the mixing driven transformation $F$ is derived as the residual of $G$ and $E$. In two-dimensional tracer space, $G$, $E$ and $F$ for each tracer surface can be derived using the inverse methods outlined in Evans et al. (2014).

In this analysis, our focus is on quantifying the processes that set the time-mean magnitude of the overturning as measured at the OSNAP mooring array (i.e. $M$). Our domain therefore encompasses the ocean north of the OSNAP mooring array, including the subpolar North Atlantic, the Nordic Seas and the Arctic Ocean. Within our analysis we do not consider the

transport through the Bering Strait, which we expect to be small (Mackay et al., 2018, 2020). We use observational-based, reanalysis-based and model-based data for $\Theta$, $S$, $q_{net}$ and $f_{net}$ to derive $G$, $E$, and $F$. In the following section we outline the specifics of these datasets.

## 2.2    Data

In the following section we will describe the datasets used to determine the drivers of the time-mean magnitude of the overturn-

ing as measured by the OSNAP array. In general, the datasets used for this analysis can be split into three groups: observational-based data, reanalyses-based data and model-based data (see Table 1). We use multiple dataset combinations within each group, giving mean estimates for $V$, $M$, $G$, $E$ and $F$ and to provide a measure of uncertainty in our final results.

To calculate $M$ for the observational dataset combinations (Table 1) we use gridded sections of $\Theta$, $S$ and $v$ from the OSNAP mooring array. The OSNAP mooring array runs from Newfoundland to the southwestern tip of Greenland and from

the southeastern tip of Greenland to Scotland (Li et al., 2021). The data spans August 2014 to May 2018, averaged into 30-day bins. We will focus on this 46 month time period for all the datasets described below, giving context to the OSNAP observations with respect to the time-mean water mass transformation.



**Table 1.** Dataset combinations for observations, reanalyses and the state estimate, showing the datasets used for $\Theta$ and $S$, $q_{net}$ and $f_{net}$, and $M$

|  | $\Theta, S$ | $q_{net}, f_{net}$ | $M$ |
|---|---|---|---|
| Observations | EN4 ARMOR3D | ERA5 NCEP CFSv2 JRA55 | OSNAP |
| Reanalyses | GLORYS2V4 ORAS5 GloSea5 C-GLORSv7 | ERA5 | GLORYS2V4 ORAS5 GloSea5 C-GLORSv7 |
| State Estimate | ECCOv4r4 | ECCOv4r4 | ECCOv4r4 |

To calculate $V$ and derive $G$ for the observational dataset combinations we use EN4 and ARMOR3D. EN4 is an objective analysis of subsurface $\Theta$ and $S$ profiles. Here we use EN4 version EN.4.2.1 with the bias corrections .g10, downloaded 2019-10-24 (Good et al., 2013). EN4 has a monthly temporal resolution and 1° by 1° horizontal resolution, with 42 irregularly spaced depth levels. ARMOR3D is a multi-observational analysis from which we use three-dimensional fields for $\Theta$ and $S$ (Guinehut et al., 2012). We access the monthly means of the reprocessed data, with a horizontal resolution of 0.25° by 0.25° and 50 irregularly spaced vertical levels. For both EN4 and ARMOR3D we use data that spans the same time period as the OSNAP data.

For the calculation of $E$, we use ERA5, NCEP-CFSv2 and JRA55. From ERA5 we use surface latent heat flux, surface sensible heat flux, surface net solar radiation, surface net thermal radiation, evaporation and total precipitation, all at a monthly temporal resolution and a 0.25° by 0.25° horizontal resolution (Hersbach et al., 2021). From NCEP-CFSv2 we use downward longwave radiation flux, upward longwave radiation flux, downward shortwave radiation flux, upward shortwave radiation flux, sensible heat flux, latent heat flux and evaporation minus precipitation (Saha et al., 2012). We use monthly-mean NCEP-CFSv2 data with a 0.5° by 0.5° horizontal resolution. From JRA55 we accessed the monthly mean model resolution two-dimensional average diagnostic fields for latent heat flux, sensible heat flux, downward longwave radiation flux, upward longwave radiation flux, downward solar radiation flux, upward solar radiation flux, evaporation and total precipitation (Japan Meteorological Agency, 2013). All fields have a horizontal resolution of $\sim 0.5° \times 0.5°$.

For the reanalysis dataset combinations we use four reanalyses distributed as part of the CMEMS Global Ocean Ensemble Reanalysis: GLORYS2V4 from Mercator Ocean (Fr), ORAS5 from ECWMF, GloSea5 from Met Office (UK), and C-GLORSv7 from CMCC (It). Each have three-dimensional fields for $\Theta$, $S$ and velocity at a 0.25° by 0.25° horizontal resolution and with 75 irregularly spaced depth levels. We use this reanalysis-based data to calculate $V$, $M$ and $G$, and we use ERA5 to calculate $E$ in order to derive $F$. Finally, for the model-based analysis we use the Estimating the Circulation and Climate of the Ocean version 4 release 4 (ECCOv4r4) state estimate to calculate $V$, $M$, $G$, $E$ and $F$. From ECCOv4r4 we use three-





dimensional fields for $\Theta$, $S$, velocity, $q_{net}$ and $f_{net}$ (Forget et al., 2015; Fukumori et al., 2021a, b). We use monthly fields at a horizontal resolution of $1°$ by $1°$. While $f_{net}$ in ECCOv4r4 incorporates freshwater fluxes due to ice and runoff, the atmospheric reanalyses datasets (ERA5, NCEP-CFSv2 and JRA55) we use here, on the other hand, do not. However, this does not appear to affect our conclusions, which are consistent across dataset combinations.

For EN4 and ARMOR3D we use the Thermodynamic Equation of State 2010 (TEOS-10; IOC et al., 2010) to calculate
conservative temperature, absolute salinity and potential density. For the reanalysis datasets and ECCOv4r4 we use the 1980 International Equation of State (EOS-80) to calculate potential density. Throughout we continue to refer to conservative temperature and model-based temperature as $\Theta$, and absolute salinity and model-based salinity as $S$. In addition, we use potential density anomaly ($\sigma$) where $\sigma = \rho - 1000\,\mathrm{kg\,m^{-3}}$. We use the full temporal range of ECCOv4r4 (1992-2018), allowing us to represent the variability of the time-mean by showing the range associated with the standard deviation of the annual means.

In summary, we create three dataset combinations, combining observational-based data, reanalysis-based data and model-based data. We use these dataset combinations to determine drivers of the time-mean magnitude of the overturning as measured at the OSNAP mooring array. For each dataset combination, we calculate the volume of water $V$ within $\Theta$, $S$, $\sigma$ and $(\Theta, S)$ classes for the regions of the North Atlantic Ocean and Arctic Ocean north of the OSNAP mooring array. From $V$ and the observed or modelled transport across the OSNAP array (i.e. $M$) we derive the transformation $G$. When calculating volume
change (Eq. 1) we use a central differencing scheme. Using observed and modelled air–sea buoyancy fluxes we calculate the air–sea driven transformation $E$. Following Eq. (4), we use $G$ and $E$ to derive the residual (mixing driven) transformation $F$. This allows us to compare the mean magnitude of the overturning streamfunction at OSNAP (i.e $M$) to the drivers of water mass transformation (i.e. $E$ and $F$) to the north of the OSNAP mooring array.

## 3    Results

We will now examine the time-mean magnitude of the overturning as measured at the OSNAP mooring array, and compare to the water mass transformation due air–sea fluxes and mixing. We will begin by describing the distribution of water mass volume within $(\Theta, S)$ space. We will then discuss the time-mean overturning in $\sigma$, $\Theta$, $S$ and $(\Theta, S)$ space. Finally, we will analyse the geographic location of the water mass transformation due to air–sea fluxes and mixing, determining the geographical regions that are critical to setting the strength of the overturning circulation at OSNAP.

### 3.1    The distribution of water masses in $(\Theta, S)$ space

The water masses of the subpolar North Atlantic are generally aligned according to their temperature (Fig. 1). In Fig. 1(a) we define some of the key water masses in the subpolar North Atlantic and Nordic Seas as a guide. For more detailed definitions see also Fig. 2 in Mackay et al. (2020). Working from warmest to coldest, the warm, salty and light waters that flow north into the subpolar North Atlantic fall along a broad mode of high volume from $12°C$ to $6°C$ (box 1 Fig. 1(a)). The gradual
cooling and freshening of water as it transits through the subpolar basin is manifest as a tilt in this mode of high volume. At $\Theta < 6°C$, this voluminous mode splits as indicated by the red arrows in Fig. 1(a). This represents a geographical separation of





water, where the fresher mode represents the water that continues to circulate within the subpolar basin, and the saltier mode indicating water that continues northward into the Nordic Seas.

The distribution of LSW and overflow water are indicated by boxes 2 and 3 in Fig. 1(a), although there is likely some

overlap between boxes. Generally LSW is fresher than overflow water. While overflow water is typically colder than LSW, some overflow water classes can have a similar temperature to LSW. A notable characteristic of the coldest overflow waters is the distinct mode of very high volume indicated by box 3 that stretches towards the Arctic and Nordic Sea dense waters indicated by box 4 in Fig. 1(a). The water that occupies the coldest and freshest quadrant of $(\Theta, S)$ space is the Arctic surface water.

The curvature of the isopycnals represented by the dotted lines in Fig. 1 emphasise the separation between the regions where temperature or salinity control density stratification. In Fig. 1 this separation is apparent at $\Theta \sim 2\,°\mathrm{C}$. At $\Theta > 2\,°\mathrm{C}$, the volumetric distribution falls within a narrow range of $S$, suggesting temperature dominates density stratification. Where $\Theta < 2\,°\mathrm{C}$, the volumetric distribution spans a much larger range in $S$, indicating the dominant role for salinity in setting the density stratification.

### 3.2    Time-mean overturning

### 3.2.1    $\sigma$ space

We will now examine the overturning streamfunction in $\sigma$ space represented here by $M(\sigma^*)$. The magnitude of the time-mean diapycnal overturning streamfunction as measured by the OSNAP mooring array has a maximum of 15.02 Sv at $\sigma = 27.675\,\mathrm{kg\,m^{-3}}$ (Fig. 2(a)). This is slightly less than the value given in Li et al. (2021) for the same dataset. Here we present

the maximum of the time-mean diapycnal overturning streamfunction as opposed to the time-mean of the maximum diapycnal overturning streamfunction at each time-step shown in Li et al. (2021). We also interpolate the OSNAP data onto the same time-step as EN4/ARMOR3D, which shortens the time-series slightly and lowers the time-mean overturning. The magnitude of the time-mean diapycnal overturning streamfunction in the reanalysis dataset combination is larger than the observations with a value of 20.36 Sv at a higher density of $\sigma = 27.725\,\mathrm{kg\,m^{-3}}$ (Fig. 2(b)). In contrast, the diapycnal overturning streamfunction

in ECCOv4r4 is lower than both OSNAP and the reanalyses at 13.16 Sv, but the density at the maximum overturning is the same as OSNAP (Fig. 2(c)).

In all datasets, $M(\sigma^*)$ represents a predominantly negative diapycnal transformation between $\sim 27.25\,\mathrm{kg\,m^{-3}}$ and $\sim 27.8\,\mathrm{kg\,m^{-3}}$, while it is typically zero at other values of $\sigma$. This represents the implied diapycnal transformation associated with warm/salty/light water entering the subpolar basin, and cold/fresh/dense water leaving the subpolar basin. It also repre-

sents the water mass transformation by air-sea fluxes and mixing that must occur within our domain to convert the incoming warm/salty/light water in the outgoing cold/fresh/dense water. As $M(\sigma^*)$ is negative in this framework, the minimum value of $M(\sigma^*)$ therefore represents the overturning strength.

The remaining terms in Figs. 2(a)-(c) represent the total diapycnal transformation (i.e. from Eq. 1: $\int \frac{\mathrm{d}V(C^*,t)}{\mathrm{d}t}\,\mathrm{d}C$), the air–sea flux driven diapycnal transformation $E(\sigma^*)$ and the residual or mixing driven diapycnal transformation $F(\sigma^*)$. By





construction, the sum of $E(\sigma^*)$, $F(\sigma^*)$ and $M(\sigma^*)$ should equal the total transformation. If the mean total transformation is equal to zero, then $-M = E + F$. Otherwise if the total transformation is non-zero, the residual transformation required to explain $M(\sigma^*)$, with respect to the value of $E(\sigma^*)$, is the difference between $F(\sigma^*)$ and the total transformation.

    At the density of minimum $M(\sigma^*)$ (i.e. the overturning strength), we can therefore examine the extent to which $M(\sigma^*)$ is explained by $E(\sigma^*)$, and thus establish the importance of $F(\sigma^*)$ in setting the maximum magnitude of $M(\sigma^*)$. In both the

observations and ECCOv4r4, $E(\sigma^*)$ accounts for most of the magnitude of $M(\sigma^*)$ at the density of minimum $M(\sigma^*)$. In the observations, a residual transformation of $F = 3.82$ Sv is required to explain the magnitude of $M(\sigma^*)$, whereas the required residual transformation in ECCOv4r4 is much smaller at -0.01 Sv. Conversely, in the reanalyses, $E(\sigma^*)$ is much smaller at the density of minimum $M(\sigma^*)$, and therefore a larger residual transformation is required balance the larger magnitude of $M(\sigma^*)$, giving $F = 15.69$ Sv.

Broadening our perspective to the full range of $\sigma$ shown in Fig. 2, we see that in all the dataset combinations $E(\sigma^*)$ always drives a positive diapycnal transformation, making water at all $\sigma$ classes more dense. This positive diapycnal transformation peaks at 12.65 Sv ($\sigma = 27.525\,\mathrm{kg\,m^{-3}}$), 4.77 Sv ($\sigma = 27.675\,\mathrm{kg\,m^{-3}}$) and 15.83 Sv ($\sigma = 27.525\,\mathrm{kg\,m^{-3}}$) in the observations, the reanalyses and ECCOv4r4 respectively. Therefore in all dataset combinations, the maximum time-mean diapycnal transformation by air–sea fluxes is always at a lighter density than the maximum magnitude of $M(\sigma^*)$.

In both the observations and ECCOv4r4, the transformation by $E(\sigma^*)$ is compensated by a negative residual diapycnal transformation (i.e $F(\sigma^*)$) in the lightest density classes, particularly where the magnitude of $M(\sigma^*)$ is small or close to zero. In the observations and ECCOv4r4, $F(\sigma^*)$ becomes less negative up to the density of minimum $M(\sigma^*)$. In the observations, $F(\sigma^*)$ becomes positive, implying densification, around the density of maximum $E(\sigma^*)$, reaching a maximum of 11.38 Sv at $27.775\,\mathrm{kg\,m^{-3}}$. In ECCOv4r4, $F(\sigma^*)$ becomes positive at approximately the density of minimum $M(\sigma^*)$ reaching a maximum

of 2.08 Sv also at $27.775\,\mathrm{kg\,m^{-3}}$. In the densest classes, $F(\sigma^*)$ is negative in both the observations and ECCOv4r4, suggesting mixing leads to a lightening, reaching a minimum of -6.95 Sv at $27.975\,\mathrm{kg\,m^{-3}}$ and -8.16 Sv at $27.875\,\mathrm{kg\,m^{-3}}$ respectively. In the reanalyses, the weaker air–sea flux transformations and the stronger overturning leads to a consistently positive $F(\sigma^*)$ at $\sigma < 27.825\,\mathrm{kg\,m^{-3}}$, reaching a maximum of 12.60 Sv at the density of minimum $M(\sigma^*)$. At $\sigma > 27.825\,\mathrm{kg\,m^{-3}}$ $F(\sigma^*)$ is negative, reaching a minimum of -7.00 Sv at $27.875\,\mathrm{kg\,m^{-3}}$.

That $E(\sigma^*)$ is always positive, and peaks at a density that is less than the density at the maximum magnitude of $M(\sigma^*)$, suggests that air–sea fluxes play key role in preconditioning the water masses that are exported across the OSNAP section, ultimately forming water within the subpolar mode water class. Thus, air–sea fluxes densify the warm/salty/light waters that enter the subpolar North Atlantic across the OSNAP section, but form water that is predominantly lighter than the water that is eventually exported southward.

Acting in concert with the uniform densification by $E(\sigma^*)$, the change in sign of $F(\sigma^*)$ at $\sigma > 27.5\,\mathrm{kg\,m^{-3}}$ implies that mixing leads to a convergence of volume in the water mass classes associated with LSW and the overflow waters. We see that $F$ drives a densification of water at densities just greater than the density of the maximum magnitude of $M(\sigma^*)$, and a lightening at the highest densities, leading to a convergence that forms the cold/fresh/dense water exported south across the OSNAP section.



In general, air-sea fluxes appear to precondition the warm/salty/light water imported into the subpolar basin, while mixing appears to modulate the eventual properties of the cold/fresh/dense waters exported southward across the OSNAP section. Here, we also see how the subtle difference between the datasets affect the relative roles of air–sea fluxes and mixing in setting the strength of the overturning in density space. For example, a weaker overturning in ECCOv4r4 means that less mixing is required to account for the overturning, while stronger overturning and weaker air-sea fluxes in the reanalyses means that more

mixing is required to drive the stronger diapycnal transformation. Furthermore, it is notable that more transformation by mixing occurs at the densest classes in the observations compared to ECCOv4r4 and the reanalyses, which coincides with the fact that less overflow water is exported southward across the OSNAP section in ECCOv4r4 and the reanalyses.

### 3.2.2   $\Theta$ and $S$ space

The diapycnal overturning streamfunction and the associated transformations due to air-sea fluxes and mixing can be separated
into its diathermal and diahaline components (Figs. 3 and 4), revealing the importance of $\Theta$ and $S$ changes in setting density variability. In both $\Theta$ and $S$ space, the diathermal and diahaline overturning streamfunction ($M(\Theta^*)$ and $M(S^*)$ respectively) are both positive, where warm and salty water enters the subpolar basin, and cold and fresh water leaves the basin. $M(\Theta^*)$ reaches a maximum of 21.08 Sv at 4.00 °C in the observational dataset combination (Fig. 3(a)), compared to a maximum of 24.08 Sv at 4.25 °C in the reanalyses (Fig. 3(b)), and 16.79 Sv at 4.00 °C in ECCOv4r4 (Fig. 3(c)). $M(S^*)$ reaches a maximum
of 15.77 Sv at 35.17 g/kg $\mathrm{g\,kg^{-1}}$ in the observational dataset combination (Fig. 4(a)), compared to a maximum of 18.29 Sv at 34.92 in the reanalyses (Fig. 4(b)), and 14.68 Sv at 35.02 in ECCOv4r4 (Fig. 4(c)). The diathermal overturning strength is always larger than the diapycnal overturning streamfunction in each dataset combination, whereas the diahaline overturning streamfunction is larger in the observations and ECCOv4r4.

   In all dataset combinations, at the $\Theta$ of maximum $M(\Theta^*)$, $E(\Theta^*)$ accounts for a smaller proportion of $M(\Theta^*)$ than for the
equivalent transformations in $\sigma$ space. In the observations, $E(\Theta^*)$ contributes -9.63 Sv at the $\Theta$ of maximum $M(\Theta^*)$, requiring a transformation of -13.25 Sv by $F(\Theta^*)$. The ratio of $E(\Theta^*)$ to $F(\Theta^*)$ at the $\Theta$ of maximum $M(\Theta^*)$ is similar in ECCOv4r4, at -7.33 Sv and -11.47 Sv respectively. In the reanalyses, while the overturning is stronger, the air–sea flux transformation at the $\Theta$ of maximum $M(\Theta^*)$ is weaker (-4.01 Sv) compared to the the observations and ECCOv4r4, therefore $F(\Theta^*)$ is larger at -21.04 Sv. In $S$ space, $E(S^*)$ is weak in all dataset combinations, and $F(S^*)$ is therefore required to explain almost all of
the overturning streamfunction at the $S$ of maximum $M(S^*)$.

   Typically, the diathermal transformation driven by air-sea heat fluxes ($E(\Theta^*)$) is much larger than the diahaline transformation due to air-sea freshwater fluxes ($E(S^*)$) at all values of $\Theta$ and $S$, suggesting that in the subpolar North Atlantic and the Nordic Seas, air–sea heat fluxes dominate the diapycnal transformation by air–sea fluxes. At all temperatures, $E(\Theta^*)$ is generally negative, implying a cooling. In the observations and the reanalyses there is no obvious peak in this transformation,
however most of the transformation by air–sea heat fluxes affect water warmer than the $\Theta$ of maximum $M(\Theta^*)$. In contrast, there are two clear peaks in $E(\Theta^*)$ in ECCOv4r4 at $\sim 3.25°C$ and at $\sim -1.5°C$. As in $\sigma$ space, a negative $E(\Theta^*)$ at all temperatures suggests a role for air–sea fluxes in the preconditioning the warm and salty water advected northward into the subpolar basin, but not in the formation of the water masses exported southward across the OSNAP section. Again, the varia-





tion in $F(\Theta^*)$ with respect to $\Theta$ highlights that mixing plays a key role in the eventual formation of the water masses exported

southward from the subpolar basin. In all datasets, we see a convergence of water mass volume driven by $F(\Theta^*)$ at the temperatures associated with LSW and the overflow waters, with a negative transformation at warmer temperatures, and positive transformation at colder temperatures.

### 3.2.3    $(\Theta, S)$ space

In the previous sections we described the time-mean overturning in $\sigma$, $\Theta$ and $S$ space. This highlighted the role of air–sea

buoyancy fluxes in forcing a cooling-driven densification of the warm, salty and light water that enters the subpolar North Atlantic and Nordic Seas. However, this cooling and densification predominantly forms subpolar mode water classes at a $\sigma$ and $\Theta$ that is lighter and warmer than the $\sigma$ and $\Theta$ of maximum overturning at the OSNAP section. Instead, water mass transformation by mixing leads to the formation of the cold, dense and fresh water exported across the OSNAP section. This formation is linked to the convergence of a mixing-driven cooling and densification in subpolar mode water classes, and a

mixing-driven warming and lightening in the coldest and densest water mass classes. In this section, we will focus on the time-mean transformation in $(\Theta, S)$ space.

In $(\Theta, S)$ space, diathermal and diahaline transformations are shown as a vector representing the thermohaline water mass transformation (Figs. 5-7). In Figs. 5-7 a vertical vector indicates a diathermal transformation and change in temperature, while a horizontal vector indicates a diahaline transformation and a change in salinity. The transformation by air–sea fluxes in the

observations (Fig. 5b) the reanalyses (Fig. 6b) and ECCOv4r4 (Fig. 7b) again highlights the role of air–sea heat fluxes in preconditioning the warm, salty and light Atlantic water that enters the subpolar North Atlantic across the OSNAP section. The bulk of the diathermal air–sea flux transformation acts on water that is warmer and lighter than the $\Theta$ and $\sigma$ of maximum $|M|$. The exception is ECCOv4r4, in which air–sea fluxes drive a large transformation in the LSW classes near 3.5°C. Generally, the diahaline component of the air–sea flux transformation is small, except near the freezing point of sea water where sea ice

formation and melting leads to a strong diahaline transformation.

The transformation implied by the transport of water across the OSNAP section has both diathermal and diahaline contributions in all dataset combinations (Figs. 5d, 6d and 7d) linked to the $\Theta/S$ contrast between the imported warm, salty and light Atlantic waters and the exported cold, fresh and dense NADW (note that the densest classes of NADW are absent in ECCOv4r4). This highlights that in the absence of a strong diahaline transformation by air–sea freshwater fluxes, the residual

transformation must account for the freshening of the Atlantic waters imported northward across the OSNAP section. This freshening is clearly evident in the negative diahaline residual transformations in water mass classes warmer than the $\Theta$ of maximum $M(\Theta^*)$ (Figs. 5c, 6c and 7c), where the residual transformation also drives a cooling. We see a warming and salinification by the residual transformation in the Arctic and Nordic Sea water masses, which when combined with the cooling and freshening in the Atlantic waters, leads to the mixing-driven convergence of volume in the NADW classes discussed in the

previous section.

An important point to note here is the angle with which isopycnals intersect the volumetric distribution and diapycnal water mass transformation in $(\Theta, S)$ space. As a result, the diapycnal transformations in $\sigma$ space (i.e. integrated along isopycnals)





merge those diapycnal transformations in warm/salty and cold/fresh water masses along a given isopycnal (Fig. 1). This will likely conflate diapycnal transformation, for example, in the North Atlantic Waters with diapycnal transformation in the high

Arctic, therefore confusing the interpretation of our analysis in $\sigma$ space.

### 3.3 Geographical distribution of water mass transformation by air–sea fluxes and mixing

In the previous sections we discussed the time mean overturning streamfunction in the context of the water mass transformation by air–sea fluxes and mixing within $\sigma$, $\Theta$, $S$ and $(\Theta, S)$ space. In this section we will discuss the geographical distribution of time-mean diapycnal and diathermal water mass transformations over subpolar North Atlantic and Nordic Seas.

Diapycnal and diathermal water mass transformations are remapped from tracer space into geographical space. At each time step and within each individual dataset, the range of tracer values for each tracer bin (i.e. $C^* \pm \Delta C/2$) are located in the three-dimensional geographical tracer field. The value of the water mass transformation within the given tracer bin is then assigned to each point in the three-dimensional geographical tracer field with the value $C^* \pm \Delta C/2$. The full depth-mean values are then calculated for the respective dataset combinations. The air–sea flux transformation is mapped to the surface tracer distribution

only, and is therefore not averaged with respect to depth.

Further, to emphasise the key geographical regions linked to the water mass transformation by air–sea fluxes and mixing, we select a series of tracer bands in $\sigma$ and $(\Theta, S)$ space linked to the key drivers of overturning discussed above. For $\sigma$ space we select 27.4 $\mathrm{kg\,m^{-3}}$ to 27.6 $\mathrm{kg\,m^{-3}}$, 27.6 $\mathrm{kg\,m^{-3}}$ to 27.77 $\mathrm{kg\,m^{-3}}$, and 27.8 $\mathrm{kg\,m^{-3}}$ to 28 $\mathrm{kg\,m^{-3}}$. For $(\Theta, S)$ space we select all values of $S$ and the $\Theta$ bands 5°C to 7.5°C, 3.5°C to 5°C, and -1°C to 2.5°C. For the reanalysis dataset combination

we extend the coldest $\Theta$ band to -1.5°C to capture the full range of residual transformation in this dataset. Remapping the diathermal transformation from $(\Theta, S)$ bins limits potential overlap between dynamically distinct regions in $(\Theta, S)$ that would otherwise be merged when remapping the diathermal transformations from $\Theta$ space. This conflation of diapycnal water mass transformation between regions of contrasting $\Theta/S$, but with similar $\sigma$, is a complication within $\sigma$ space.

The lightest and warmest of these bands select the $\sigma$ and $\Theta$ range in which air–sea fluxes cool the warm, salty and light water

imported northwards across the OSNAP section. The central bands focus on the $\sigma$ and $\Theta$ of maximum $|M|$, where the residual transformation generally drives the largest cooling and densification. The final bands select the $\sigma$ and $\Theta$ corresponding to the mixing-driven warming and lightening within the densest water masses of the Nordic and Arctic Seas.

### 3.3.1 27.4 $\mathrm{kg\,m^{-3}}$ to 27.6 $\mathrm{kg\,m^{-3}}$

In the observational dataset combination, the densification by air–sea fluxes in the 27.4 $\mathrm{kg\,m^{-3}}$ to 27.6 $\mathrm{kg\,m^{-3}}$ band is largest

within the Iceland Basin, Irminger Sea and Labrador Sea (Fig. 8a). Within the Nordic Seas this densification is also elevated along the path of Atlantic Water in the Norwegian Sea and along the coast of East Greenland. Within this density range the time-mean residual diapycnal transformation is positive. The remapped residual transformation has a distinctive pattern with a positive diapycnal transformation in the Norwegian Sea adjacent to a region of negative transformation in the Greenland Sea, and general positive transformation throughout the Iceland Basin, Irminger Sea and Labrador Sea (Fig. 8b). This pattern

of positive and negative transformation in the Norwegian and Greenland Seas is an artefact of a non-uniformly distributed



seasonal cycle, where summer time lightening in the Greenland Sea exceeds the wintertime densification in the same region, while in the Norwegian Sea wintertime densification exceeds the summertime lightening.

Reflecting the contrast in the relative strength of the air–sea flux and residual transformations in the observations and reanalyses (Fig 2), the remapped transformations in the $27.4 \, \mathrm{kg\,m^{-3}}$ to $27.6 \, \mathrm{kg\,m^{-3}}$ band for the reanalysis dataset combination highlights the weak densification by air–sea fluxes and a stronger densification implied by the residual transformation (Figs. 9a and 9b). Similar to the observations, the air–sea flux and residual transformations in the reanalysis dataset combination are largest in the Iceland Basin, Irminger Sea, Labrador Sea and Norwegian Sea. However there is no positive diapycnal transformation by air–sea fluxes along the coast of East Greenland in the reanalyses. In addition, the same pattern of positive and negative residual diapycnal transformation is present in the Norwegian and Greenland Seas associated with the seasonal cycle in these regions.

The remapped diapycnal transformation by air–sea buoyancy fluxes for ECCOv4r4 in the $27.4 \, \mathrm{kg\,m^{-3}}$ to $27.6 \, \mathrm{kg\,m^{-3}}$ band is similar to both the observations and reanalyses (Fig. 10a). The residual transformation, on the other hand, is different, with a predominently negative transformation throughout the subpolar basin and the Nordic Seas (Fig. 10b). This reflects the values of the time-mean residual transformation within this $\sigma$ range in $\sigma$ space (Fig 2c). This negative transformation is weakest in the Nordic Seas, where the air–sea flux transformation is highest.

### 3.3.2 $27.6 \, \mathrm{kg\,m^{-3}}$ to $27.77 \, \mathrm{kg\,m^{-3}}$

In the central density band, that selects for the diapycnal transformation near the density of maximum $|M(\sigma^*)|$, the remapped diapycnal transformation by air–sea fluxes are restricted to a smaller geographical area indicative of the distribution of isopycnal outcrops in this range of $\sigma$. In the observations (Fig. 8c) and ECCOv4r4 (Fig. 10c), the transformation by air–sea buoyancy fluxes is largest in the Irminger Sea and western Labrador Sea, with no diapycnal densification in the Iceland Basin. The densification by air–sea fluxes is also elevated along the coasts of Norway and East Greenland in both the observations and ECCOv4r4, while the diapycnal water mass transformation is also high in the Norwegian Sea in the observations. As expected, the remapped diapycnal transformations by air–sea fluxes in the reanalyses are weaker, and the geographical distribution is also more restricted (Fig. 9c). Similar to the observations and ECCOv4r4, densification by air–sea fluxes in the reanalyses is highest in the Irminger Sea, Labrador Sea and Norwegian Sea. However, there is no diapycnal transformation along the coasts of Norway and East Greenland.

There is better agreement between the remapped residual diapycnal transformations in the observations and reanalyses for the $27.6 \, \mathrm{kg\,m^{-3}}$ to $27.77 \, \mathrm{kg\,m^{-3}}$ band (Figs. 8d and 9d). On the other hand, the remapped residual transformation in ECCOv4r4 agrees less well with the observations and reanalyses within this $\sigma$ band (Fig. 10d). In both the observations and reanalyses, the remapped residual transformation is highest in the Irminger Sea, Labrador Sea and Norwegian Sea. The region of negative residual transformation in the Greenland Sea is again an artefact of a geographically non-uniform seasonal cycle. While the residual transformation is generally weaker in ECCOv4r4, densification is also elevated to a lesser extent in the Irminger Sea, Labrador Sea and Norwegian Sea.





### 3.3.3  27.8 kg m$^{-3}$ to 28 kg m$^{-3}$

Within the densest $\sigma$ band, the remapped diapycnal air–sea flux transformation in all dataset combinations is generally strongest in the central Nordic Seas, where transformation in the lighter $\sigma$ bands is weaker (Figs. 8e, 9e and 10e), highlighting that the densest isopycnals outcrop in this region. The remapped residual transformation for the 27.8 kg m$^{-3}$ to 28 kg m$^{-3}$ band is generally confined to the Nordic Seas in the observations (Fig. 8f) and reanalyses ((Fig. 9f). The sign of the residual transformation in the Nordic Seas implies a mixture of densification and lightening in the observations and general lightening in

the reanalyses. In ECCOv4r4, the remapped residual transformation extends beyond the Nordic Seas into the western Iceland Basin, the western Irminger Sea and the western Labrador Sea (Fig. 10).

In each dataset combination, the difference in the residual transformation between the 27.8 kg m$^{-3}$ to 28 kg m$^{-3}$ $\sigma$ and 27.6 kg m$^{-3}$ to 27.77 kg m$^{-3}$ bands imply a convergence of volume within the density range of NADW. While the distribution of the remapped residual transformation in the observations and reanalyses suggests a mixing-driven lightening of dense water in the

Nordic Seas, only the remapped residual transformation in ECCOv4r4 implies a mixing-driven lightening along the pathways of ISOW and DSOW in the Iceland Basin and Irminger Sea respectively, and in the Labrador Sea. However, that the diapycnal transformations shown here compound transformations from dynamically distinct regions (i.e. contrasting spiciness) some geographical detail may be lost in the process of remapping the transformations. This may be less of an issue for diathermal transformation remapped from $(\Theta, S)$ space.

### 3.3.4  5°C to 7.5°C


With the remapped diathermal transformations, there is generally a better agreement between the dataset combinations compared to the remapped diapycnal transformations. In the warmest $\Theta$ band, corresponding to the $\Theta$ range of subpolar mode water, the remapped diathermal transformations by air–sea heat fluxes indicate a cooling along the pathways of Atlantic Water in the Iceland Basin, the Irminger Sea, and the Nordic Seas (Figs. 11a, 12a and 13a). The northern extent of this cooling follows

the strong sea surface $\Theta$ (SST) gradient between 3°C and 7°C, running from the Irminger Sea to the north east through the Nordic Seas. A transition to diathermal warming north of this frontal region indicates the northward migration of isothermal outcrops during the summer. Further, within this temperature range, air–sea heat fluxes do not drive a cooling in the Labrador Sea, contrary to the equivalent diapycnal transformation in $\sigma$ space. The remapped diathermal transformations by air–sea heat fluxes are strongest in ECCOv4r4 and weakest in the reanalyses.

The remapped residual diathermal transformation in the 5°C to 7.5°C band implies a mixing driven cooling everywhere except the Greenland Sea (Figs. 11b, 12b and 13b). Typically this mixing driven cooling is strongest in the frontal region of the Irminger Sea and Nordic Seas. In all dataset combinations, the residual diathermal transformation also implies a mixing-driven cooling in the Labrador Sea.





### 3.3.5 3.5°C to 5°C

Near the $\Theta$ of maximum $M(\Theta^*)$, the remapped diathermal transformation by air–sea heat fluxes drives a cooling that is confined to a narrower geographical region constrained by the cooler isotherms along the strong frontal region discussed above (Figs. 11c, 12c and 13c). This cooling does not impact the Iceland Basin or the eastern Irminger Sea, and additionally drives a cooling in the Labrador Sea. Consistent with the warmer $\Theta$ band, the remapped cooling by air–sea heat fluxes is strongest in ECCOv4r4 and weakest in the reanalysis dataset combination.

In all dataset combinations, the residual diathermal transformation is largest in the 3.5°C to 5°C band (Figs. 11d, 12d and 13d). This mixing-driven cooling is greatest in the Nordic Seas, corresponding to the strong frontal region. Further, mixing leads to large cooling in the Iceland Basin and Irminger Sea, particularly over the Reykjanes Ridge. Additionally, within this temperature band, mixing cools water in the Labrador Basin.

### 3.3.6 -1°C to 2.5°C

In the coldest temperature band, the remapped diathermal transformations by air–sea heat fluxes are shifted northward across the frontal region in the Nordic Seas (Figs. 11e, 12e and 13e). The influence of air–heat fluxes is reduced in the Irminger Sea, and is confined to the coastal region along the southeastern tip of Greenland, where air–sea heat fluxes drive a cooling, particularly in the observations and ECCOv4r4. In addition, within the -1°C to 2.5°C band, air–sea heat fluxes affect the Labrador Sea differently in each dataset combination. In the observations, air–sea heat fluxes drive a cooling around the

boundary of the Labrador Sea, implying that isotherms in the -1°C to 2.5°C band don't outcrop in the interior Labrador Sea. In the reanalyses, the time-mean air–sea heat fluxes are weak in the Labrador Sea within the -1°C to 2.5°C band. Contrastingly, air–sea heat fluxes in ECCOv4r4 drive a strong cooling throughout the Labrador Sea in the -1°C to 2.5°C band.

The remapped residual diathermal transformations in the -1°C to 2.5°C band imply a mixing-driven warming in all dataset combinations (Figs. 11f, 12f and 13f). In the observations, this warming is evident across the Nordic Seas, and is particularly

strong along the Iceland-Scotland ridge and along the southeastern tip of Greenland. Further, this mixing-driven warming is strongest in the reanalysis dataset combination, particularly within the Nordic Seas and the Labrador Sea. Within ECCOv4r4, the residual warming is uniform across the Nordic Seas and along the southeastern tip of Greenland, but is highest in the Labrador Sea.

## 4 Discussion and Conclusions

In this study, we've used a combination of observational-based, reanalysis-based and model-based data for temperature, salinity, air–sea buoyancy fluxes and meridional transport to quantify the drivers of the diabatic overturning in the subpolar North Atlantic and Nordic Seas. We have established the first observational based, basin wide estimate of mixing in the subpolar North Atlantic, and for the first time quantified the role of mixing in setting the strength of the diapycnal overturning. Using a water mass transformation framework we compare the overturning streamfunction in $\sigma$, $\Theta$, $S$ and $(\Theta, S)$ space at the OSNAP





mooring array to the diabatic transformation by air–sea buoyancy fluxes and mixing in the regions to the north of the OSNAP
mooring array. Our analysis focused on a comparison between the time-mean overturning streamfunction and the diabatic
transformation by air–sea fluxes and mixing over the time period of the OSNAP observations (August 2014 to May 2018).

We show that the time-mean overturning streamfunction is set by both mixing and air–sea buoyancy fluxes. The diapycnal
and diathermal transformation by air–sea fluxes is largest at densities and temperatures lighter and warmer than the density of

the maximum overturning at OSNAP. This indicates that air–sea heat fluxes precondition the warm and salty Atlantic Water that
circulates around the subpolar North Atlantic and Nordic Seas, contributing to the formation of subpolar mode water classes
within density classes lighter than the density of maximum overturning at OSNAP. Notably, air–sea fluxes typically account for
a weaker freshening than implied by the contrast between the incoming salty Atlantic Water and outgoing fresh North Atlantic
Deep Water (Le Bras et al., 2021).

Mixing plays two key roles in setting the properties of the water masses that participate in the subpolar overturning. Firstly,
in the absence of a substantial freshening by air–sea freshwater fluxes in the subpolar North Atlantic and Nordic Seas, mixing
freshens Atlantic Water to form subpolar mode water in concert with air–sea heat fluxes. Further complimenting the precon-
ditioning by air–sea heat fluxes, mixing is ultimately responsible for the formation of the North Atlantic Deep Water classes
exported southward across the OSNAP section. A densification by mixing near the density of maximum overturning, and a

lightening in the coldest and densest classes, implies a convergence of volume in North Atlantic Deep water classes, where
mixing acts on the preconditioned subpolar mode water, recently convected water in the subpolar basins, and on the dense
waters formed in the Nordic and Arctic Seas.

Remapping the water mass transformation by air–sea fluxes and mixing to geographical space highlights the key regions
of water mass transformation in the subpolar North Atlantic and Nordic Seas. The preconditioning of warm, salty and light

Atlantic water typically occurs along the pathways of Atlantic Water, identified by the strong lateral SST gradients in the
Nordic Seas and the Iceland Basin and Irminger Sea. In addition, this cooling by air–sea heat fluxes is matched by a mixing
driven cooling along a similar frontal region. In the Nordic seas, the cooling by air–sea fluxes and mixing will progressively
contribute to the densification of water leading to the formation of the dense waters that fill the Nordic Seas. In contrast, the
mixing driven cooling within the Iceland Basin and Irminger Sea may be linked to the exchange between the cooler/fresher

water of the basin interiors and the warmer/saltier water of the shallower boundary currents.

Near the $\sigma$ and $\Theta$ of maximum $|M|$, air–sea heat fluxes drive a cooling that is confined to a narrower geographical region
constrained by slightly cooler isotherms along the same SST front associated with the pathway of Atlantic Water in the Nordic
Seas. Further, in this $\sigma$ and $\Theta$ range, air–sea fluxes do not affect the Iceland Basin or the eastern Irminger Sea, with the strongest
cooling in the western Irminger Sea and the central Labrador Sea, likely linked to convection in these regions. Within the same

$\sigma$ and $\Theta$ range, mixing drives a cooling over most of the Nordic Seas, Irminger Sea, Iceland Basin, and Labrador Sea. Mixing
is particularly elevated along the SST front in the Nordic Seas and over the Reykjanes Ridge.

Within the Irminger Sea, Iceland Basin, and the Labrador Sea, the mixing driven cooling near the temperature and density
of maximum overturning is likely linked to the exchange between the basin interiors and the boundary currents. This exchange





would occur between the densest mode waters, recently convected water in the Labrador Sea and Irminger Sea interiors, and
ISOW and DSOW in the deep boundary currents of the subpolar basins. This processes subsequently forms upper NADW.

Generally, the influence of air-sea fluxes is further reduced within the coldest and densest classes. In addition, within this
$\sigma$ and $\Theta$ range, mixing in the Nordic seas, over the Greenland-Scotland Ridge and along the southeastern coast of Greenland
drives a warming and lightening. This is linked to the transformation of the densest water masses as they circulate within the
Nordic Seas and along the East Greenland current, progressively warming and lightening to eventually form lower NADW.

This study emphasises the role of mixing in setting the magnitude of the overturning streamfunction, and highlights how
mixing works together with air–sea fluxes to transform Atlantic Water into North Atlantic Deep Water. This allows us to
contextualise what we know about mixing processes and the dynamics within the subpolar basins and Nordic Seas in terms of
the diapycnal overturning.

For instance, the role of air–sea fluxes in the formation of subpolar mode water is well established. Petit et al. (2020) and Petit
et al. (2021), for example, show that air–sea buoyancy fluxes are important for the diapycnal water mass transformation within
the density range associated with subpolar mode water. Here, we also emphasise the importance of mixing in the formation
of subpolar mode water, where mixing drives a freshening of Atlantic Water. The source of this freshwater is ultimately fresh
Arctic water that flows into the subpolar basin along the East Greenland Current, or via Baffin Bay and into the Labrador Sea
(Håvik et al., 2017; Le Bras et al., 2018; Foukal et al., 2020; Holliday et al., 2020). Mixing between the fresher Arctic-sourced
waters and Atlantic Water would then likely occur within the boundary currents of the subpolar basins (Pennelly et al., 2019).

Considering the formation of the source waters of the Greenland-Scotland Ridge overflows, Mauritzen (1996) show that
Atlantic Waters entering the Nordic Seas loose sufficient buoyancy to be classified as overflow water before reaching the
Barents Sea or Fram Strait. This dense water eventually flows out of the Nordic Seas as overflow waters in the East Greenland
Current via either the Arctic or as so called return Atlantic Water (Våge et al., 2013; Bower et al., 2019). They also highlight
that intermediate water formation in the interior of the Greenland and Iceland Seas contribute to the water that overflows the
Iceland-Scotland Ridge as observed by de Jong et al. (2018). Our results suggest that mixing plays a key role in the densification
of Atlantic Waters, both along the pathways of Atlantic Water northward through the Nordic Seas and within the interior of the
Greenland and Iceland Seas. The proximity of the mixing-driven cooling of Atlantic Water to the SST front within the Nordic
Seas highlights the potential importance of turbulence associated with eddy-driven isopycnal mixing across the SST front (Lee
et al., 1997; Marshall and Speer, 2012) or submesocale processes leading to the cross-front exchange of potential vorticity
anomalies (Thomas et al., 2013).

The processes that govern the mixing-driven water mass transformations that form North Atlantic Deep Water are well
described in the literature. The mixing driven cooling inferred in the boundary currents of the subpolar basins are described
in idealised models (Spall and Pickart, 2001; Spall, 2003, 2004; Straneo, 2006; Brüggemann and Katsman, 2019) to infer a
diapycnal overturning. Further, exchange between recently convected water in the interior of the Irminger Sea and the East
Greenland current was inferred by Le Bras et al. (2020), leading to an along isopycnal homogenisation of temperature and
salinity properties. Turbulent mixing within the deep overflow waters downstream of the Greenland-Scotland Ridge has been
documented by many (e.g. Fer et al., 2010; Beaird et al., 2012; North et al., 2018). The mixing-driven transformation we infer





here is therefore consistent with observations and simulations of turbulent mixing in the subpolar North Atlantic and Nordic
Seas.

Our results highlight some important considerations for the application of different dataset combinations in the context of
understanding the diapycnal drivers of subpolar overturning. For example, at the resolution of the model-based reanalyses and
ECCOv4r4, the narrow boundary currents of the subpolar North Atlantic and Nordic Seas will not be resolved. This will have
consequences for the mixing driven water mass transformation in these regions, and affect the model's representation of the
diabatic overturning. This is particularly evident in the absence of mixing driven cooling around the boundary of the Iceland
Basin and the Irminger Sea within the reanalysis dataset combination, for example. Further, in ECCOv4r4, the southward
export of overflow waters are not realistically represented at the OSNAP section, highlighting a deficiency in the representation
of either overflow water formation, or the in the dynamics affecting the overflow waters in their pathways around the subpolar
basins to the OSNAP section. In addition, the simulation of large heat loss in the Labrador Sea at the $\sigma$ and $\Theta$ of maximum
overturning within ECCOv4r4 compared to observations implies unrealistic convection within ECCOv4r4, a common issue
within ocean models in this region as addressed by Yeager et al. (2021).

There are also notable differences between the dataset combinations with respect to the relative strengths of the overturning,
the air–sea flux driven transformation and the mixing driven transformation. For example, mixing near the density of maximum
overturning is weaker in ECCOv4r4 compared to the observations and reanalyses. In contrast, the overturning strength is similar
in the observations and ECCOv4r4, but stronger in the reanalyses. Further, the reanalyses have a weaker air-sea flux driven
transformation and the strongest mixing driven transformation. As such, ECCOv4r4 potentially gets the right answer (correct
overturning) for the wrong reasons (weak mixing, stronger air-sea fluxes). Whereas the reanalyses gets the wrong answer
(strong overturning), for the right reasons (stronger mixing).

In context of understanding the drivers of the diapycnal overturning at OSNAP, our analysis highlights the challenges asso-
ciated with focusing solely on the role of air–sea fluxes in driving the transformation at the density of maximum overturning.
Notably this approach misses the fact that air–sea fluxes predominantly precondition Atlantic Water to form subpolar mode
water, and that mixing is ultimately responsible for setting the properties of the North Atlantic Deep Water classes exported
southward across the OSNAP section. Further, analysis of the water mass transformation at the density of maximum overturn-
ing doesn't necessarily give any information about the water that is exported southward in the overturning circulation. This is
linked to the fact that the isopycnal associated with the density of maximum overturning has a broad geographical distribution
with a large range in spiciness. So that the formation of water denser than the isopycnal of maximum overturning doesn't nec-
essarily produce water with the $(\Theta, S)$ characteristics of North Atlantic Deep Water. By considering the full $(\Theta, S)$ water mass
transformation budget, we can show the complete picture of the processes that lead to the transformation of Atlantic Water and
the formation of North Atlantic Deep Water.

Finally, that mixing is so critical for the formation of North Atlantic Deep Water, highlights the need to characterise and
quantify the regional dynamics and processes responsible for the generation of turbulent mixing in the subpolar North At-
lantic and Nordic Seas with targeted observations, and to assess the relative importance of these processes for the diapycnal
overturning. The ultimate goal would be to better inform model parameterisations of these processes, leading to an improved





representation of the overturning circulation in climate models, and better predictions of the response of the overturning circu-
lation to anthropogenic climate change and its impact on heat and carbon uptake.

*Data availability.*  EN.4.2.1 data were obtained from https://www.metoffice.gov.uk/hadobs/en4/ and are Crown Copyright, Met Office, 2021,
provided under a Non-Commercial Government Licence http://www.nationalarchives.gov.uk/doc/non-commercial-government-licence/version/
2/. ARMOR3D can be downloaded from the Copernicus Marine Service under the product identifier MULTIOBS_GLO_PHY_TSUV_
3D_MYNRT_015_012 at https://resources.marine.copernicus.eu/?option=com_csw&view=details&product_id=MULTIOBS_GLO_PHY_
TSUV_3D_MYNRT_015_012. ERA5 data were accessed from the Copernicus Data Store: https://doi.org/10.24381/cds.f17050d7. NCEP-
CFSv2 data were accessed from the NCAR/UCAR Research Data Archive: https://doi.org/10.5065/D69021ZF. JRA55 data were accessed
from the NCAR/UCAR Research Data Archive https://doi.org/10.5065/D60G3H5B. The CMEMS Global Ocean Ensemble Reanalysis
can be downloaded from the Copernicus Marine Service under the product identifier GLOBAL_REANALYSIS_PHY_001_031 at https://
resources.marine.copernicus.eu/?option=com_csw&view=details&product_id=GLOBAL_REANALYSIS_PHY_001_031. ECCOv4r4 was
accessed via https://ecco-group.org/. OSNAP data are available via https://www.o-snap.org/data-access/

*Author contributions.*  DGE conceptualised the study and conducted the investigation with feedback from NPH, SB and IAALB. DGE
prepared the manuscript with contributions from all co-authors.

*Competing interests.*  The authors declare that they have no conflict of interest.

*Acknowledgements.*  DGE, NPH and SB were funded under the NERC research grant NE/R015953/1. IAALB was funded by the NSF grant
OCE-2038481.





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





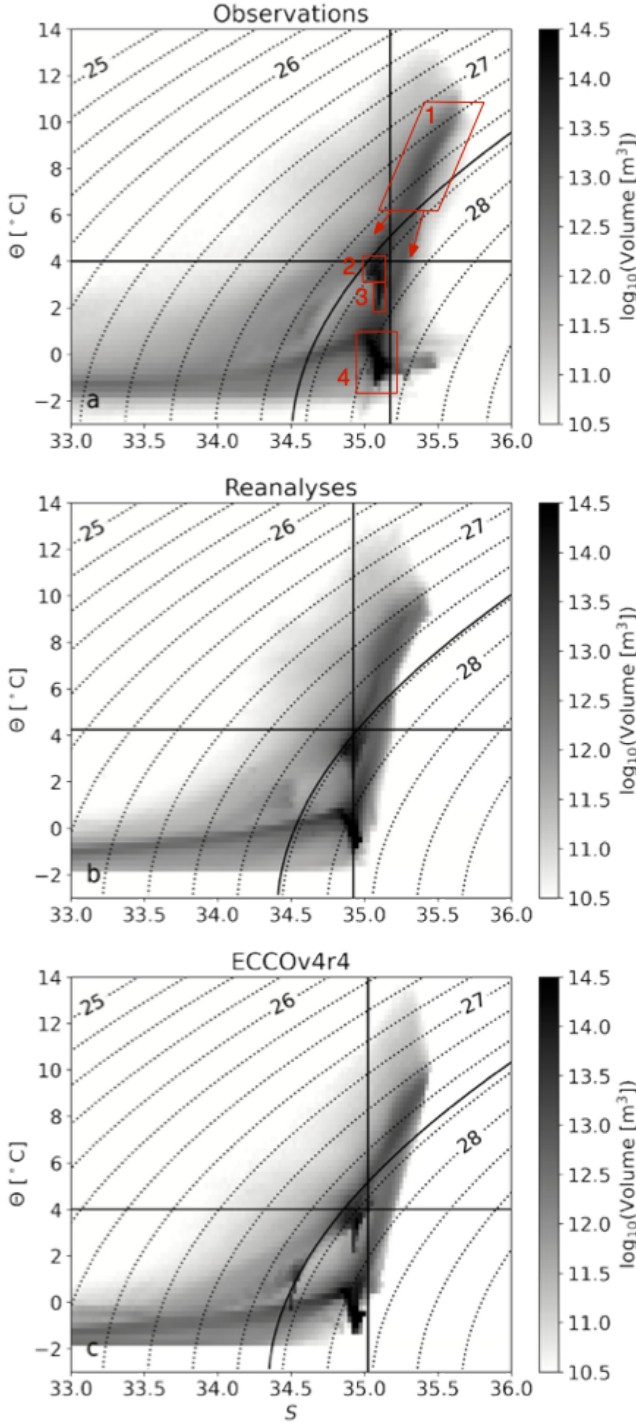

**Figure 1.** Time-mean volumetric distribution of the region to the north of the OSNAP section for (a) the observational dataset combination, (b) the renanlysis dataset combination and (c) ECCOv4r4. The curved dotted contours show $\sigma$ with units of $\mathrm{kg\,m^{-3}}$. The solid contour, horizontal line and vertical line represent the $\sigma$, $\Theta$ and $S$ of maximum overturning respectively. The red boxes broadly define the key water masses in the region to the north of the OSNAP section: 1. North Atlantic Water, 2. LSW and OW, 3. OW, 4. Nordic Sea and Arctic Deep water. Also of note is that the 28 $\mathrm{kg\,m^{-3}}$ isopycnal approximately marks the density at the maximum depth of the Greenland-Scotland ridge.





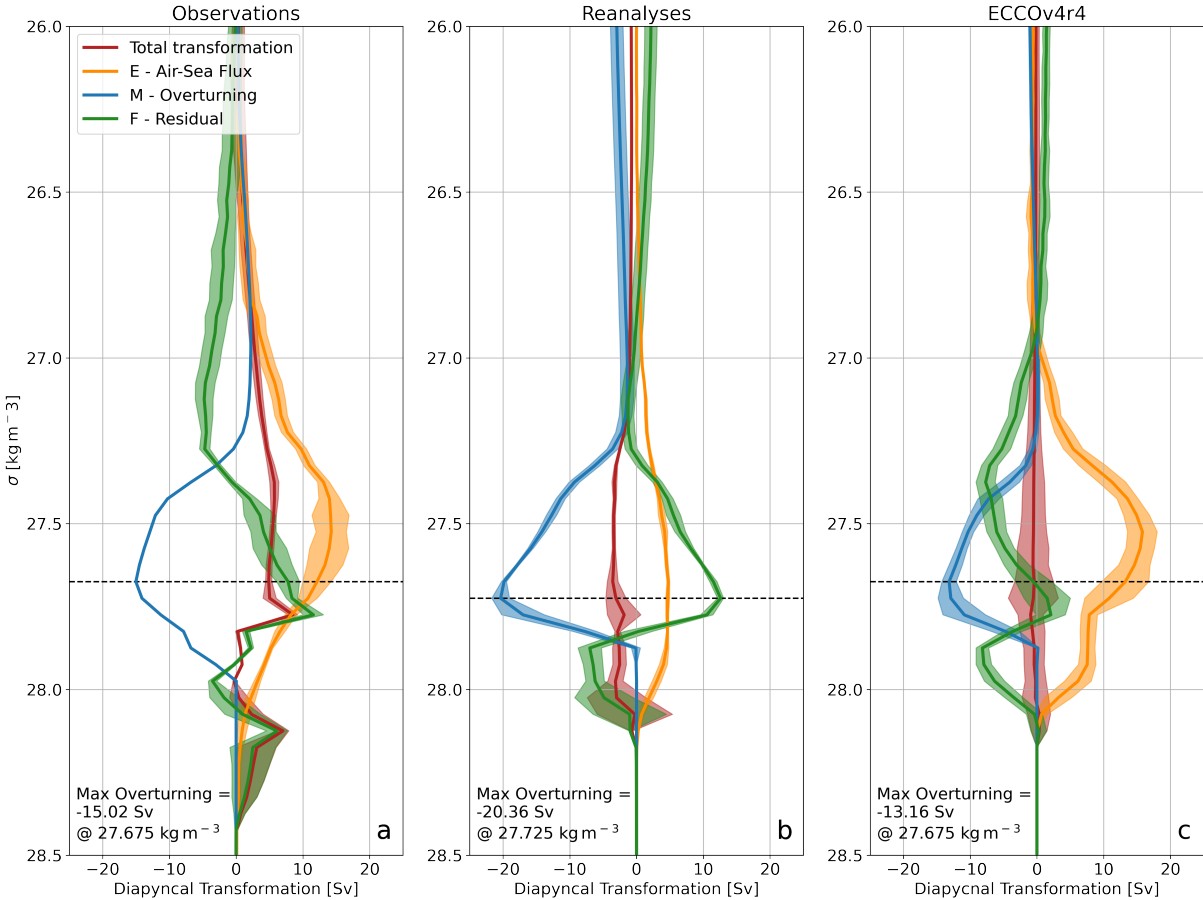

**Figure 2.** Time-mean diapycnal water mass transformation of the region to the north of the OSNAP section for (a) the observational dataset combination, (b) the renanalysis dataset combination and (c) ECCOv4r4. The panels show the total transformation (red), the air–sea flux transformation (orange), the overturning streamfunction (blue) and the residual transformation (green). The shading in panels (a) and (b) represent the standard deviation of the respective dataset combinations. The shading in panel (c) represents the standard deviation of the annual means in ECCOv4r4.



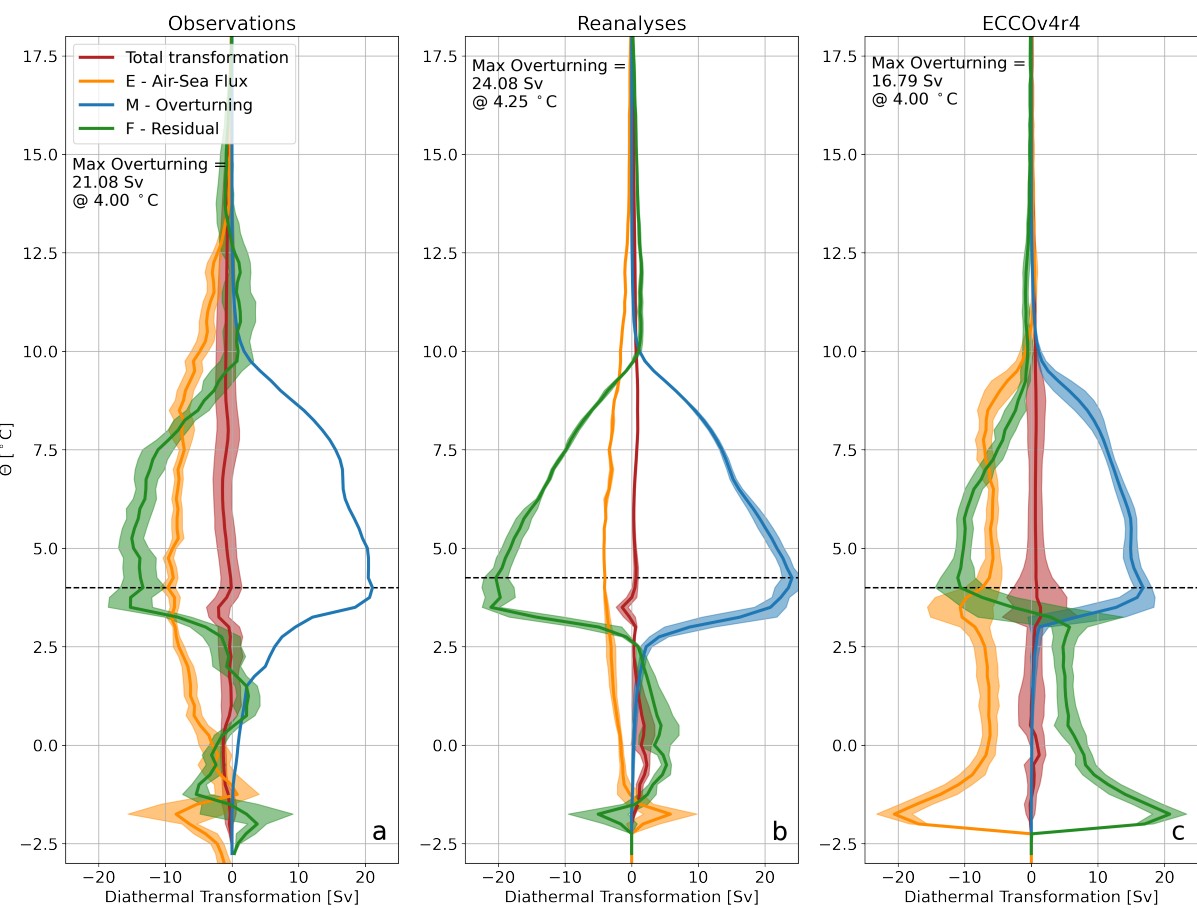

**Figure 3.** As in Fig. 2 but for the diathermal water mass transformation.





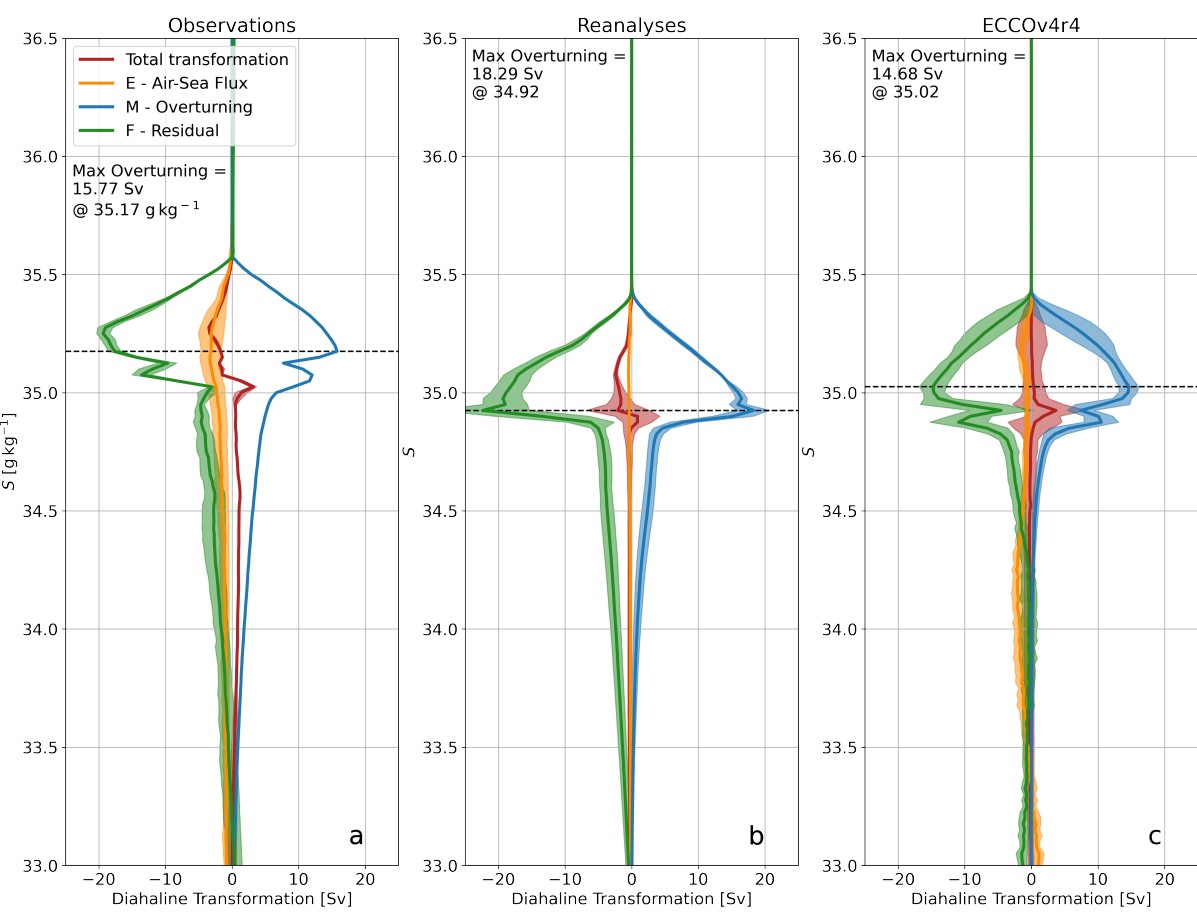

**Figure 4.** As in Fig. 2 but for the diahaline water mass transformation.





**Figure 5.** Time-mean thermohaline water mass transformation of the region to the north of the OSNAP section for the observational dataset combination showing (a) the total transformation, (b) the air–sea flux transformation, (c) the residual transformation and (d) the overturning streamfunction. The vectors represent the combined diahaline (horizontal component) and diahaline (vertical component) transformations (units: Sv), while the colours show the volume change associated with each term. The grey contours in (a) show the time-mean volumetric distribution as shown in Fig. 1. The solid black contours show isopycnals at $25\,\mathrm{kg\,m^{-3}}$, $26\,\mathrm{kg\,m^{-3}}$, $27\,\mathrm{kg\,m^{-3}}$ and $28\,\mathrm{kg\,m^{-3}}$. The dashed black contour, horizontal line and vertical line represent the $\sigma$, $\Theta$ and $S$ of maximum overturning respectively.





**Figure 6.** As in Fig. 5 but for the reanalysis dataset combination.





**Figure 7.** As in Fig. 5 but for ECCOv4r4.





**Figure 8.** Air–sea flux and residual diapycnal water mass transformation remapped from $\sigma$ space into geographical space for the observational dataset combination. The values represent the time- and depth-mean remapped transformation. Each row shows a different $\sigma$ range: (a)-(b) 27.4 kg m$^{-3}$ to 27.6 kg m$^{-3}$, (c)-(d) 27.6 kg m$^{-3}$ to 27.77 kg m$^{-3}$, and (e)-(f) 27.8 kg m$^{-3}$ to 28 kg m$^{-3}$. The grey contours show the mean surface $\sigma$. LS = Labrador Sea, IS = Irminger Sea, IB = Iceland Basin, NS = Nordic Seas, and *RR* = Reykjanes Ridge.





**Figure 9.** As in Fig. 8 but for the reanalysis dataset combination.

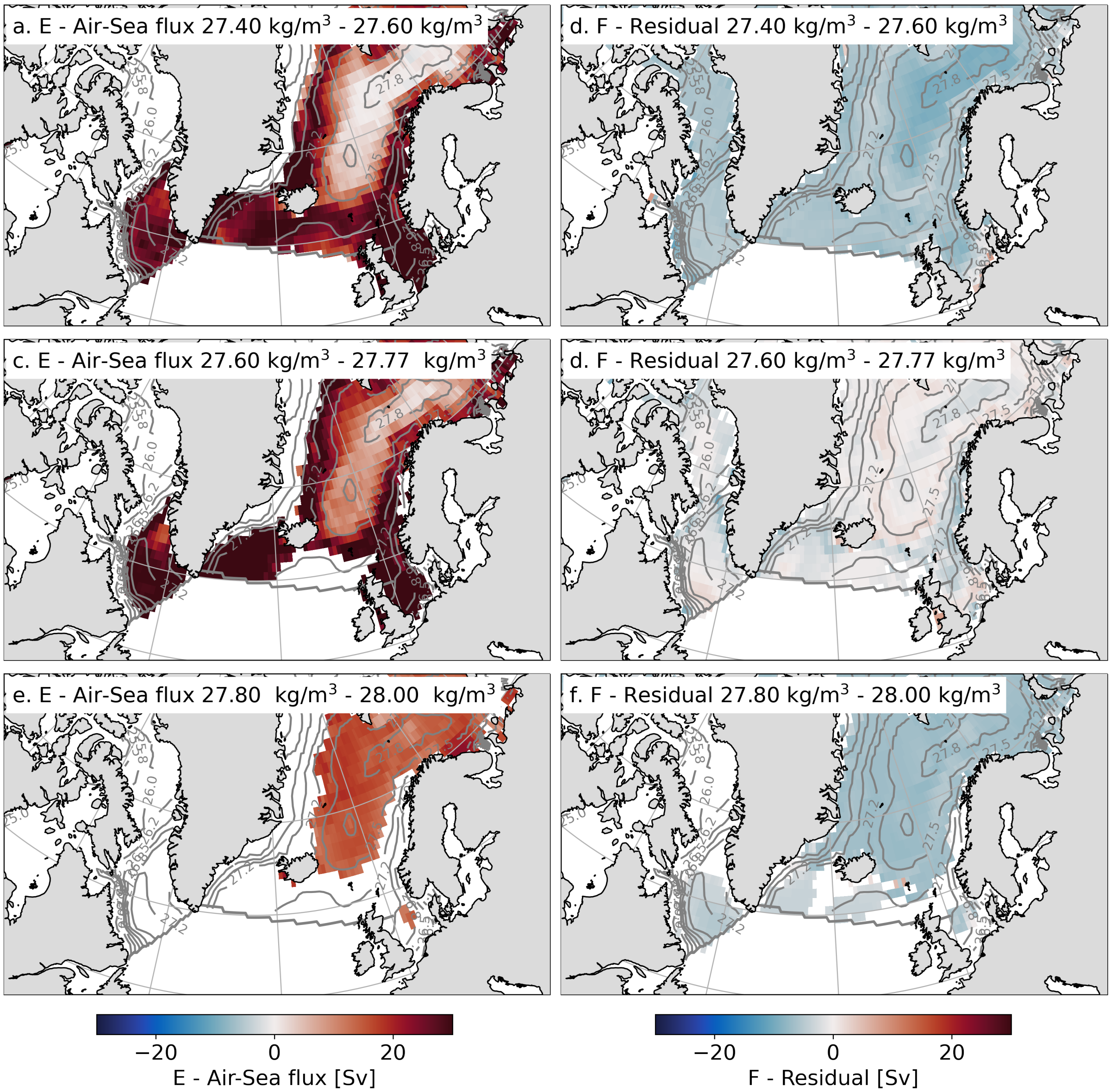

**Figure 10.** As in Fig. 8 but for ECCOv4r4.





**Figure 11.** Air–sea flux and residual diathermal water mass transformation remapped from $(\Theta, S)$ space into geographical space for the observational dataset combination. The values represent the time- and depth-mean remapped transformation. Each row shows a different $\Theta$ range: (a)-(b) 5°C to 7.5°C, (c)-(d) 3.5°C to 5°C, and (e)-(f) -1°C to 2.5°C. The grey contours show the mean surface $\Theta$.





**Figure 12.** As in Fig. 11 but for the reanalysis dataset combination. Note panels (e) and (f) show -1.5°C to 2.5°C





**Figure 13.** As in Fig. 13 but for ECCOv4r4.