# Peer review of "Mixing and air-sea buoyancy fluxes set the time-mean overturning circulation in the subpolar North Atlantic and Nordic Seas"

_EGUsphere, 2022_

## Referee Comment (RC1)

egusphere-2022-1059

**Mixing and air-sea buoyancy fluxes set the time-mean overturning circulation in the subpolar North Atlantic**

by Dafydd Gwyn Evans, N. Penny Holliday, Sheldon Bacon, and Isabela Le Bras

In this manuscript the water mass transformation by air-sea buoyancy fluxes and mixing in the subpolar North Atlantic and Nordic Seas is quantified. Observational, reanalysis, and numerical data are used in a comprehensive water mass transformation framework to determine where and how the dense water masses that supply the lower limb of the Atlantic Meridional Overturning Circulation at the OSNAP mooring array are formed.

I think this is a well-written and very interesting manuscript that significantly contributes to the understanding of water mass transformation and overturning in the subpolar North Atlantic and the Nordic Seas, in particular by emphasizing the importance of mixing. I have only a few comments and concerns, primarily related to uncertainties in the estimates and the occasional need for clarification elaborated on below, that I hope the authors will consider.

**General comments:**

The inability of coarse-resolution products such as the reanalyses and ECCO state estimate to resolve boundary currents is discussed. This consideration applies also to both of the observational data sets. Furthermore, the scarcity of data on the east Greenland shelf (e.g., Behrendt *et al.*, 2018) implies that the gridded products here primarily result from extrapolation unconstrained by direct observations. While the East Greenland Current is becoming ice-free in winter (Moore *et al.*, 2022), the same cannot be said for the shelf, so I am surprised to see air-sea interaction occurring along most or all of east Greenland despite the presence of sea ice in Figure 8a. I think these issues require more extensive consideration.

What is the magnitude of the error in your estimates? The mixing, in particular, is equated with the residual of the total and flux-driven transformations, and probably has substantial uncertainty.

I think some clarification regarding water mass products and the regional distribution of water masses is necessary. I concur that cooling by air-sea heat fluxes and freshening by mixing precondition the warm and salty Atlantic Water as it progresses northward. In the Iceland Basin and in the Irminger Sea this leads to the formation of subpolar mode water. In the Nordic Seas overflow water is formed, not subpolar mode water, which can be misunderstood from the abstract (line 8) and text (line 471). In the present climate deep or bottom water formation occurs at most to a very limited extent in the Nordic Seas (line 37). The dense water formed in the Greenland Sea, where the deepest and densest convection occurs, is not too dense to cross the Greenland-Scotland Ridge and contribute to the overflows (Brakstad *et al.*, 2019; Huang *et al.*, 2020). It is also not entirely clear in Figure 1 what you refer to as overflow water. Boxes 2 and 3 are too warm to represent the pure overflow water that crosses the Greenland-Scotland Ridge (see Figure 6 in Smedsrud *et al.*, 2022), so I suppose it must be the DSOW and NEADW found south of the ridge after mixing with and entrainment of subpolar water masses? In my opinion, adding a map to show the geographical distribution of the water masses in Figure 1 would have provided important clarification.

The water mass transformation framework is probably not familiar to most readers. I think that the narrative would benefit from occasionally summarizing key findings and recasting the explanations using more general terms, such as at the end of the data and methods section. Perhaps a schematic illustration would also be helpful.

I think that some sort of integrated measure of water mass transformation would have been very useful to include. Petit *et al.* (2020) found that the Iceland Basin and Irminger Sea were particularly important sources of dense water to the lower limb of the AMOC, but they did not account for mixing. How do your results reinforce or modify those of Petit *et al.* (2020) if you quantify the total transformation south and north of the Greenland-Scotland Ridge, considering also the preconditioning that occurs as the Atlantic Water flows northward and the regions of enhanced mixing downstream of the overflows across the Greenland-Scotland Ridge?

In this manuscript the time-mean overturning circulation is considered, while the water mass transformation by buoyancy loss from air-sea interaction occurs in winter. How would the results differ if you consider the water mass transformation that occurs in winter only? Without the seasonal cycle, Figure 8 would have been easier to interpret.

**Specific comments:**

Line 21:
There is substantial difference in density between LSW and the overflow waters formed in the Nordic Seas (e.g., Pacini *et al.*, 2020; Mastropole *et al.*, 2017; Smedsrud *et al.*, 2022).

Line 37:
In the present climate, only intermediate water masses are formed by gradual transformation of Atlantic Water (Mauritzen, 1996) and by open-ocean convection in the Greenland and Iceland Seas (Huang *et al.*, 2020). These intermediate water masses can contribute to the overflows across the Greenland-Scotland Ridge and are not confined to the Nordic Seas.

Line 135:
The transport through Bering Strait is on the order of 1 Sv (Woodgate, 2018). If you mean that this is negligible, you should write that instead.

Line 155:
How come a combination of ERA5, NCEP, and JRA55 is used for the fluxes in the observations, while only ERA5 is used for the fluxes in the reanalyses? This should be justified, in particular as the observational- and reanalysis-based results may be more directly compared if the only difference between the estimates are the oceanic parameters.

Line 205:
More importantly for the vertical structure of the subpolar North Atlantic, LSW is less dense than overflow water (e.g., Pacini *et al.*, 2020).

Lines 218 and elsewhere:
An estimate of the uncertainties, both in the magnitude of the overturning and in the density of maximum overturning, would have been very valuable to include. Without error bars, it is impossible to know how much confidence to place in these estimates.

Line 276:
Downstream of the overflows, where the mass transport approximately doubles by entrainment of ambient water masses (Dickson and Brown, 1994; Girton and Sanford, 2003), must be key locations for the mixing that leads to a convergence of volume within the density range of NADW. As discussed, the overflows are not well represented in the reanalyses and state estimate, but most likely not in the observational products either due to

the coarse resolution. What are the implications for your estimates of mixing?

Line 353:
How come the upper limit for the middle density class is 27.77 kg/m$^3$ rather than 27.8 kg/m$^3$?

Line 369 and elsewhere:
The region of negative transformation extends into the Iceland Sea, south of the Greenland Sea. Referring to the western part of the Nordic Seas exclusively as the Greenland Sea is not correct.

Line 405:
The region of most pronounced air-sea flux transformation is primarily the western Nordic Seas, i.e., the Greenland and Iceland Seas. The Greenland Sea is also where the deepest and densest convection occurs.

Line 521:
Intermediate water masses formed in the interior Greenland and Iceland Seas also contribute to the Denmark Strait overflow (Mastropole *et al.*, 2017; Semper *et al.*, 2019).

Line 527:
This paragraph highlighting processes that govern the mixing-driven water mass transformation is important. I think some of these processes could have been elaborated on (e.g., boundary current – interior exchange) and that the overflows, which are most likely key locations for the mixing that warms and freshens the coldest and densest water masses leading to the convergence of volume within the density range of NADW, should have been emphasized.

Figure 1:
I think you need to clarify the water masses and their geographical distribution in Figure 1. In particular, what exactly do you mean by overflow water (as discussed above, the "pure" overflow water at the Greenland-Scotland Ridge is different from the water masses you identify in boxes 2 and 3) and where is the NADW? I think that a map showing the geographical distribution of these water masses would be a very valuable inclusion.

Figure 5d:
I think Figure 5d requires a bit more explanation. It is not evident why overturning implies warming and salinification.

Figure 11:
Why is the water mass transformation due to air-sea fluxes an order of magnitude lower when comparing temperature classes (Figure 11) to density classes (Figure 8)?

**Detailed comments:**

Lines 9, 207, 504, and elsewhere:
Ridge, Seas, and Current should be capitalized.

Line 84:
There's an "in" too many.

Line 84:
He would be a more appropriate pronoun.

Line 166:

It should be "...each **has**..."

Lines 430 and 432:

Inconsistent use of hyphen in "mixing-driven cooling".

**References**

Behrendt A, Sumata H, Rabe B, Schauer U. 2018. UDASH - Unified Database for Arctic and Subarctic Hydrography. *Earth System Science Data* **10**: 1119–1138, doi:10.5194/essd–10–1119–2018.

Brakstad A, Våge K, Håvik L, Moore GWK. 2019. Water mass transformation in the Greenland Sea during the period 1986-2016. *Journal of Physical Oceanography* **49**: 121–140, doi:10.1175/JPO–D–17–0273.1.

Dickson RR, Brown J. 1994. The production of North Atlantic Deep Water: Sources, rates and pathways. *Journal of Geophysical Research* **99**: 12 319–12 341, doi:10.1029/94JC00 530.

Girton JB, Sanford TB. 2003. Descent and modification of the overflow plume in the Denmark Strait. *Journal of Physical Oceanography* **33**: 1351–1364.

Huang J, Pickart RS, Huang RX, Lin P, Brakstad A, Xu F. 2020. Sources and upstream pathways of the densest overflow in the Nordic Seas. *Nature Communications* : doi:10.1038/s41 467–020–19 050–y.

Mastropole D, Pickart RS, Valdimarsson H, Våge K, Jochumsen K, Girton J. 2017. On the hydrography of Denmark Strait. *Journal of Geophysical Research: Oceans* **122**: 306–321, doi:10.1002/2016JC012 007.

Mauritzen C. 1996. Production of dense overflow waters feeding the North Atlantic across the Greenland-Scotland Ridge. Part 1: Evidence for a revised circulation scheme. *Deep Sea Research I* **43**: 769–806, doi:10.1016/0967–0637(96)00 037–4.

Moore GWK, Våge K, Pickart RS, Renfrew IA. 2022. Sea-ice retreat suggests re-organization of water mass transformation in the Nordic and Barents Seas. *Nature Communications* **13**: doi:10.1038/s41 467–021–27 641–6.

Pacini A, Pickart RS, Bahr F, Torres DJ, Ramsey AL, Holte J, Karstensen J, Oltmanns M, Straneo F, Bras IAL, Moore GWK, de Jong MF. 2020. Mean conditions and seasonality of the West Greenland Boundary Current System near Cape Farewell. *Journal of Physical Oceanography* **50**: doi:10.1175/JPO–D–20–0086.1.

Petit T, Lozier MS, Josey SA, Cunningham SA. 2020. Atlantic deep water formation occurs primarily in the Iceland Basin and Irminger Sea by local buoyancy forcing. *Geophysical Research Letters* **47**: doi:10.1029/2020GL091 028.

Semper S, Våge K, Pickart RS, Valdimarsson H, Torres DJ, Jónsson S. 2019. The emergence of the North Icelandic Jet and its evolution from northeast Iceland to Denmark Strait. *Journal of Physical Oceanography* **49**: 2499–2521, doi:10.1175/JPO–D–19–0088.1.

Smedsrud LH, Brakstad A, Madonna E, Muilwijk M, Lauvset SK, Spensberger C, Born A, Eldevik T, Drange H, Jeansson E, Li C, Olsen A, Skagseth Ø, Slater DA, Straneo F, Våge K, Årthun M. 2022. Nordic Seas heat loss, Atlantic Inflow, and Arctic sea ice cover over the last century. *Reviews of Geophysics* **60**: doi:10.1029/2020RG000 725.

Woodgate RA. 2018. Increases in the Pacific inflow to the Arctic from 1990 to 2015, and insights into seasonal trends and driving mechanisms from year-round Bering Strait mooring data. *Progress in Oceanography* **160**: 124–154, doi:10.1016/j.pocean.2017.12.007.

---

## Author Comment (AC2)

Dear Editor and Reviewers,

Thank you for taking the time to review our manuscript and for providing some very useful comments. Please find our responses to each of your comments below. Our responses are written in italic blue font. We have included a pdf showing the tracked changes in the following document:

revised_manuscript_tracked_changes.pdf

This indicates where text has been added, deleted or replaced. The line numbers indicated below refer to the tracked changes document.

**Reviewer Comment 1**

**General comments:**

The inability of coarse-resolution products such as the reanalyses and ECCO state estimate to resolve boundary currents is discussed. This consideration applies also to both of the observational data sets. Furthermore, the scarcity of data on the east Greenland shelf (e.g., Behrendt *et al.*, 2018) implies that the gridded products here primarily result from extrapolation unconstrained by direct observations.

*The reviewer is correct that these boundary currents are not resolved in the observational datasets. However, the water masses formed as a consequence of the water mass transformation in these regions are evident in the observations and should therefore be apparent in the residual transformation. In the model-based datasets, these boundary currents are not resolved and therefore the resultant mixing driven water mass transformation is not simulated, and will not be apparent in the residual transformation. We have clarified this point at lines 573-575 with the following sentence:*

*"While the dynamics associated with the boundary currents and overflows are also not resolved in the observational datasets, the products of mixing in these regions are observed and are therefore represented in the residual transformation."*

While the East Greenland Current is becoming ice-free in winter (Moore *et al.*, 2022), the same cannot be said for the shelf, so I am surprised to see air-sea interaction occurring along most or all of east Greenland despite the presence of sea ice in Figure 8a. I think these issues require more extensive consideration.

*Generally, the water mass transformations by air-sea fluxes shown in Figure 8a in the region of the East Greenland Shelf are the summertime water transformation. During the winter there is no water mass transformation by air-sea fluxes in this region. This is a consequence of the time-averaging used for these figures.*

*However, it is possible that the wintertime transformations by air-sea fluxes are shown in ice covered regions, particularly for the colder temperature classes. This is a consequence of the remapping of the transformations from temperature/salinity (T/S) space to geographical space, which remaps transformations in T/S bins to regions where T/S classes exist geographically. This, importantly, does not imply that we calculated a water mass transformation by air-sea fluxes in ice covered regions. Water mass transformation by air-sea fluxes only occurs in regions of open ocean according to the air-sea flux datasets used.*

What is the magnitude of the error in your estimates? The mixing, in particular, is equated with the residual of the total and flux-driven transformations, and probably has substantial uncertainty.

*We thank the reviewer for their suggestion, we have included an estimate of the error on our estimates of the water mass transformation within sections 3.2.1 and 3.2.2. For the observations and reanalyses this is the standard deviation of the dataset combinations. For ECCO we show the standard deviation of the annual mean.*

I think some clarification regarding water mass products and the regional distribution of water masses is necessary. I concur that cooling by air-sea heat fluxes and freshening by mixing precondition the warm and salty Atlantic Water as it progresses northward. In the Iceland Basin and in the Irminger Sea this leads to the formation of subpolar mode water. In the Nordic Seas overflow water is formed, not subpolar mode water, which can be misunderstood from the abstract (line 8) and text (line 471).

*We thank the reviewer for highlighting this potential misunderstanding, we have now clarified these sentences at lines 10 and 497*

In the present climate deep or bottom water formation occurs at most to a very limited extent in the Nordic Seas (line 37). The dense water formed in the Greenland Sea, where the deepest and densest convection occurs, is not too dense to cross the Greenland- Scotland Ridge and contribute to the overflows (Brakstad *et al.*, 2019; Huang *et al.*, 2020).

*We have removed reference to dense water in this sentence (line 40), referring only to intermediate water. We have also added the references of Brakstad et al. (2019) and Huang et al. (2020).*

It is also not entirely clear in Figure 1 what you refer to as overflow water. Boxes 2 and 3 are too warm to represent the pure overflow water that crosses the Greenland-Scotland Ridge (see Figure 6 in Smedsrud *et al.*, 2022), so I suppose it must be the DSOW and NEADW found south of the ridge after mixing with and entrainment of subpolar water masses? In my opinion, adding a map to show the geographical distribution of the water masses in Figure 1 would have provided important clarification.

*We thank the reviewer for highlighting this source for potential confusion. We have now clarified (both in the main text at line 217 and in the figure 1 caption) that we are referring to overflow waters as they exist downstream of the Greenland-Scotland Ridge. We have decided not to include an additional figure showing the geographical distribution of water masses as we want to avoid specifically defining water masses in this manuscript, and only include the definitions in figure 1 as a guide.*

The water mass transformation framework is probably not familiar to most readers. I think that the narrative would benefit from occasionally summarizing key findings and recasting the explanations using more general terms, such as at the end of the data and methods section. Perhaps a schematic illustration would also be helpful.

*We have added a paragraph to the methods section at lines 135-140 to generalise the water mass transformation framework. Rather than creating a new schematic, we instead refer to the schematics shown in Walin et al. (1982) and Evans et al. (2014).*

I think that some sort of integrated measure of water mass transformation would have been very useful to include. Petit *et al.* (2020) found that the Iceland Basin and Irminger Sea were particularly important sources of dense water to the lower limb of the AMOC, but they did not account for mixing. How do your results reinforce or modify those of Petit *et al.* (2020) if you quantify the total transformation south and north of the Greenland-Scotland Ridge, considering also the preconditioning that occurs as the Atlantic Water flows northward and the regions of enhanced mixing downstream of the overflows across the Greenland-Scotland Ridge?

*We thank the reviewer for their interesting suggestion. While an analysis of the water mass transformation between the OSNAP section and the Greenland-Scotland Ridge could be achieved using ECCO or the reanalyses, it would not be possible with the observational dataset combination given the lack of comprehensive transport measurements across the Greenland-Scotland Ridge. Indeed, in ECCO, analysis of the water mass transformation between the OSNAP section and the Greenland-Scotland Ridge reinforces the conclusions of this study, highlighting the role of air-sea fluxes in forming subpolar mode water and of mixing in driving a convergence of volume within NADW classes. However, without a comparison between the observational dataset combination and the reanalyses/ECCO, we have therefore decided not to include this additional analysis within this paper, and we will incorporate this into a future paper.*

In this manuscript the time-mean overturning circulation is considered, while the water mass transformation by buoyancy loss from air-sea interaction occurs in winter. How would the results differ if you consider the water mass transformation that occurs in winter only? Without the seasonal cycle, Figure 8 would have been easier to interpret.

*We thank the reviewer for their suggestion. However, including only the wintertime mean makes little difference for the representation of the air-sea fluxes and we feel further elaboration of the seasonal cycle is beyond the scope of this paper, which as the reviewer mentions focusses on the time-mean overturning.*

**Specific comments:**

Line 21:
There is substantial difference in density between LSW and the overflow waters formed in the Nordic Seas (e.g., Pacini *et al.*, 2020; Mastropole *et al.*, 2017; Smedsrud *et al.*, 2022).

*Here we refer to the properties of LSW and overflow water at the point at which it is exported from the Subpolar North Atlantic (i.e. southward across the OSNAP section), as opposed to their properties in the region in which they are formed. We clarified this by now stating that dense overflow waters "initially" form in the Nordic Seas. "Slightly less dense" therefore adequately describes the difference between them, especially in the context of the full water column.*

Line 37:
In the present climate, only intermediate water masses are formed by gradual transformation of Atlantic Water (Mauritzen, 1996) and by open-ocean convection in the Greenland and Iceland Seas (Huang *et al.*, 2020). These intermediate water masses can contribute to the overflows across the Greenland-Scotland Ridge and are not confined to the Nordic Seas.

*We thank the reviewer for highlighting this error. We have removed the second half of this sentence (line 40).*

Line 135:
The transport through Bering Strait is on the order of 1 Sv (Woodgate, 2018). If you mean that this is negligible, you should write that instead.

*Changed to 'negligible' as suggested (line 144).*

Line 155:
How come a combination of ERA5, NCEP, and JRA55 is used for the fluxes in the observations, while only ERA5 is used for the fluxes in the reanalyses? This should be justified, in particular as the observational- and reanalysis-based results may be more directly compared if the only difference between the estimates are the oceanic parameters.

*The reanalyses used for the reanalysis dataset combination were initialised using ERA-Interim. Therefore, we thought it appropriate to only use ERA5 as we expect the difference between ERA5 and ERA-Interim to be negligible in the context of this analysis (see the difference between ERA-Interim and NCEP-NCAR in Evans et al., 2017). We have clarified this at lines 177-180.*

Line 205:
More importantly for the vertical structure of the subpolar North Atlantic, LSW is less dense than overflow water (e.g., Pacini *et al.*, 2020).

*We have now clarified here that overflow water is denser than LSW (line 218).*

Lines 218 and elsewhere:
An estimate of the uncertainties, both in the magnitude of the overturning and in the density of maximum overturning, would have been very valuable to include. Without error bars, it is impossible to know how much confidence to place in these estimates.

*Please see our response to the reviewers previous comment on this matter.*

Line 276:
Downstream of the overflows, where the mass transport approximately doubles by entrainment of ambient water masses (Dickson and Brown, 1994; Girton and Sanford, 2003), must be key locations for the mixing that leads to a convergence of volume within the density range of NADW. As discussed, the overflows are not well represented in the reanalyses and state estimate, but most likely not in the observational products either due to the coarse resolution. What are the implications for your estimates of mixing?

*Please see our response to the reviewer's same comment above.*

Line 353:
How come the upper limit for the middle density class is 27.77 kg/m3 rather than 27.8 kg/m3?

*Here our aim was to select the density range of negative residual diapycnal transformation and 27.77 kg/m$^3$ is the density at which residual transformation crosses 0 Sv.*

Line 369 and elsewhere:
The region of negative transformation extends into the Iceland Sea, south of the Greenland Sea. Referring to the western part of the Nordic Seas exclusively as the Greenland Sea is not correct.

*We thank the reviewer pointing out this inconsistency and have corrected the manuscript accordingly at lines 393-395.*

Line 405:
The region of most pronounced air-sea flux transformation is primarily the western Nordic Seas, i.e., the Greenland and Iceland Seas. The Greenland Sea is also where the deepest and densest convection occurs.

*We have changed "central Nordic Seas" to "Greenland and Iceland Seas" (line 431).*

Line 521:
Intermediate water masses formed in the interior Greenland and Iceland Seas also contribute to the Denmark Strait overflow (Mastropole *et al.*, 2017; Semper *et al.*, 2019).

*We thank the reviewer for highlighting this issue. We have changed "Iceland-Scotland Ridge" to "Greenland-Scotland Ridge" and added the suggested references (line 546).*

Line 527:
This paragraph highlighting processes that govern the mixing-driven water mass transformation is important. I think some of these processes could have been elaborated on (e.g., boundary current – interior exchange) and that the overflows, which are most likely key locations for the mixing that warms and freshens the coldest and densest water masses leading to the convergence of volume within the density range of NADW, should have been emphasized.

*We have reordered this paragraph to emphasise the importance of mixing in overflow waters (line 553 onwards).*

Figure 1:
I think you need to clarify the water masses and their geographical distribution in Figure 1. In particular, what exactly do you mean by overflow water (as discussed above, the "pure" overflow water at the Greenland- Scotland Ridge is different from the water masses you identify in boxes 2 and 3) and where is the NADW? I think that a map showing the geographical distribution of these water masses would be a very valuable inclusion.

*Please see our response the reviewer's general comment above.*

Figure 5d:
I think Figure 5d requires a bit more explanation. It is not evident why overturning implies warming and salinification.

*We clarify this by adding the following at lines 350–353:*

*"Within the context of this framework, and the volume budget within our domain, transport of water across the OSNAP section adds warm, salty and light water to the region north of OSNAP and removes cold, fresh and dense water. The transport of water across the OSNAP line therefore warms, salinifies and lightens the region to the north of the domain, implying a positive diathermal and diahaline transformation, and a negative diapycnal transformation."*

Figure 11:
Why is the water mass transformation due to air-sea fluxes an order of magnitude lower when comparing temperature classes (Figure 11) to density classes (Figure 8)?

*For Figure 8 we remap transformations from density bins, whereas in Figure 11 we remap temperature/salinity bins. The volume in density bins is generally larger that the volume in T/S bins, so as a result the transformation between bins is larger.*

Detailed comments:

Lines 9, 207, 504, and elsewhere:
Ridge, Seas, and Current should be capitalized.

*Corrected*

Line 84:
There's an "in" too many.

*Removed*

Line 84:
He would be a more appropriate pronoun.

*Changed*

Line 166:
It should be "...each has..."

*Corrected*

Lines 430 and 432:
Inconsistent use of hyphen in "mixing-driven cooling".

*Corrected*

**Reviewer Comment 2**

Evans et al., examined overturning and water mass transformation in the region north of the OSNAP regions, from observations, reanalysis, and model. They break down the transformation in density, temperature, salinity, and temperature-salinity spaces. This type of detailed work is useful for our understanding of the water mass transformation of the region.

There are some drawbacks, however. The most important one is that there is a large difference among three products, and it is not clear how to understand them. 1. The time frame difference MAY contribute to part of the difference (and examine the result over some same full-year period would be helpful). 2. The nature of the data-assimilated products. For example, although one can simply call the residual as contribution due to mixing, it is however not necessarily due to the diapycnal mixing that is prescribed in the model. Some of the "mixing" contribution is due to the data-assimilation (which may be viewed as a "forcing" term that takes place in the full water column when there is difference between model and data profile). I am not sure what is the best way to address this issue.

*We thank the reviewer for this insightful comment, this is something we had not previously considered. While both ECCOv4r4 and the reanalyses are data-assimilated products, we believe an unphysical "forcing" term associated with the difference between model and observations will only be a potential issue with the reanalysis dataset combination. The adjoint configuration of ECCOv4r4 should not manifest in a temperature or salinity change that is not accounted for within the model's own temperature or salinity budget.*

*On the other hand, within reanalyses model output is fitted to observations in a manner that is not represented by a physical forcing term in the model. This forcing term could potentially manifest as a residual transformation in the water mass transformation framework. However, we expect this correction to act randomly with no bias (e.g. Waters et al., 2014), therefore having limited impact on the magnitude of the residual transformation.*

*Any potential impact would however be reduced by averaging the calculated water mass transformations across the reanalysis ensemble, as we do here. Quantification of this error in the residual transformation is not possible without knowing the value of the data assimilation correction, which is not available for these products.*

*We have added some discussion on this matter at lines 582-587.*

*Waters, J., Lea, D.J., Martin, M.J., Mirouze, I., Weaver, A. and While, J. (2015), Implementing a variational data assimilation system in an operational 1/4 degree global ocean model. Q.J.R. Meteorol. Soc., 141: 333-349. https://doi.org/10.1002/qj.2388*

**Details**

Title suggests "… in the subpolar North Atlantic" whereas the work clearly includes the Nordic Sea.

*We have updated the title to include the Nordic Seas.*

L6. "Complementary roles of air-sea buoyancy and mixing in setting …" because the mixing term is diagnosed as the difference between (the overturning and buoyancy forcing), it is probably more appropriate to say that the air-sea buoyancy flux alone does not account for all the transformation and mixing term can be quite significant.

*We thank the reviewer for their suggestion and we have updated this sentence as follows (line 7-8):*

*"…that air-sea fluxes alone cannot account for the time-mean magnitude of the overturning at OSNAP, and therefore a residual mixing-driven transformation is required to explain the difference."*

L12. Not sure why single out "climate models" here, all models need that. Also, I think one other key result, other than the importance of mixing which has been previously documented in numerical model, the difference among different products is large. And some discussion on this difference would be helpful.

*Here we have changed "climate models" to "ocean and climate models'.*

*As we are close to the word limit in the abstract, we have chosen to not add any additional discussion regarding the differences between the dataset combinations into the abstract.*

*However, in line with the reviewer's general comments we have expanded on the existing discussion on the differences between datasets at lines 582-587 in section 4 of the manuscript.*

L22. I guess "Bower et al. (2019)" here is referring to the spreading of NADW, but the way it is included here is a bit strange as it reads like a reference for "slightly dense water masses formed in the Irminger and Lab Seas…" even if it is for the NADW spreading, it should be noted that Bower et al. (2019) emphasized Lagrangian view, you may want to add some reference in Eulerian view as well (Rhein et al., 2015-JGR)

Rhein, M., D. Kieke, and R. Steinfeldt (2015), Advection of North Atlantic Deep Water from the Labrador Sea to the southern hemisphere, J. Geophys. Res. Oceans, 120, 2471–2487, doi:10.1002/2014JC010605

*We have added reference to Rhein et al. (2015) to the text at line 23.*

L25. There are many papers in the subpolar region, and I honestly do not expect one to include all. But I think some are important under the topic of "water mass transformation in the subpolar North Atlantic" and should be included. For example, Brambilla et al. 2008; Marsh 2000; Grist et al., 2014.

Brambilla, E., L. D. Talley, and P. E. Robbins, 2008: Subpolar mode water in the northeastern Atlantic: 2. Origin and transformation. J. Geophys. Res., 113, C04026, https://doi.org/ 10.1029/2006JC004063.

Marsh, R., 2000: Recent variability of the North Atlantic thermohaline circulation inferred from surface heat and freshwater fluxes. J. Climate, 13, 3239–3260, https://doi.org/ 10.1175/1520-0442(2000)013,3239:RVOTNA.2.0.CO;2.

Grist, J. P., S. A. Josey, R. Marsh, Y. O. Kwon, R. J. Bingham, and A. T. Blaker, 2014: The surface-forced overturning of the North Atlantic: Estimates from modern era atmospheric reanalysis datasets. J. Climate, 27, 3596–3618, https://doi.org/ 10.1175/JCLI-D-13-00070.1.

*We thank the reviewer for highlighting these missing references, we've now included them at line 28.*

The terminology is a bit strange (to me), and I assume you have followed previous study. Your total transformation is really the tendency or the drift (dv/dt), which should be close to zero if one thinks the ocean is close to a steady state. To me, the sum of overturning and the tendency should be the total transformation, which then break down into the transformation due to air-sea fluxes and due to mixing.

*We apologise for the confusion with our terminology. We have changed "total transformation" at lines 251 onwards to "volume tendency".*

Table 1. It would be useful to list the time frame considered. I assume it is the OSNAP observational period for observations, and it was mentioned that 1992-2018 is used for ECCOv4r4. But I did not find the time information for Reanalysis. Time frame difference can contribute the difference between the three products. For example, the total transformation (tendency term) is large in observations in part because it includes some seasonal variability (as OSNAP period is short and not full year), whereas this term is much smaller in ECCOv4r4 (as it is long-term and full year).

*We thank the reviewer for this suggestion. We have added the time period that we use to each of the datasets in Table 1. Further, we have experimented with different combinations of full years when calculating the time-mean, but the impact is negligible.*

L168. It is probably inappropriate to call ECCOv4r4 model based, ECCO is a state estimate which assimilates a lot of data just like reanalysis.

*We have corrected this so that we make reference to ECCO as a "state estimate" rather than using "model-based" from line 150 onwards.*

L197 and Figure 1. It is unclear the exact region "the subpolar North Atlantic and Nordic Seas" referred to. I was mentioned north of OSNAP, so it also includes the northwestern Lab Sea, the Buffin Bay (i.e., north of the OSNAP-west). On the eastern side, where is the northern boundary of Nordic Seas? Does it include the entire Arctic Ocean?

*We specify at line 196 that our domain includes everything north of the OSNAP section, including the Nordic Seas and Arctic Ocean. We have changed the text at line 210 to include the Arctic Ocean to avoid any further confusion.*

L210. Figure 1 include three panels and there seems quite some similarity and difference, none of them was mentioned in the manuscript.

*We have added some discussion of the differences between the datasets at lines 228-232.*

L285. 35.17 g/kg I guess?

*Corrected*

Diathermal transformation in Figure 3, why the air-sea flux term differs so much, especially in ECCO there is large transformation close to -2C, is that due to ice? In general, working on temperature and salinity space is quite dangerous for large area, because water masses with very different density and source may end up at similar temperature/salinity. And it is quite difficult to understand the difference between Figures 2-4 (do you change signs? That the overturning is on positive side in Figure 2 but on negative side in Figures 3-4)

*The reviewer is correct that the large transformation in ECCOv4r4 at temperatures of -2˚C are associated with ice formation/melt. The transformations associated with ice formation/melting are not accounted for in the observations or the reanalysis. However, this discrepancy does not seem to affect our conclusions.*

*With regards to using density, temperature or salinity space, we note in section 3.1 that much of the volume of water in the Subpolar North Atlantic and Nordic Seas ($\Theta > 1˚C$) is more closely oriented according to temperature, and we see that there is a large degree of overlap between geographically distinct regions along isopycnals with very different spiciness.*

*Further, following from Zou et al. (2020) we see that focusing solely on water mass transformations in density space masks significant diathermal and diahaline water mass transformations that are likely important to understand, for example, why models typically simulate a larger than observed overturning in the Labrador Sea.*

*The sign of the overturning streamfunction changes between figures 2-4 as within the context of this framework, and the volume budget within our domain, transport of water across the OSNAP section adds warm, salty and light water to the region north of OSNAP and removes cold, fresh and dense water. The transport of water across the OSNAP line therefore warms, salinifies and lightens the region to the north of the domain, implying a positive diathermal and diahaline transformation, and a negative diapycnal transformation. We have clarified this at lines 350-353.*

---

## Referee Report (RR1)

egusphere-2022-1059

**Mixing and air-sea buoyancy fluxes set the time-mean overturning circulation in the subpolar North Atlantic**

by Dafydd Gwyn Evans, N. Penny Holliday, Sheldon Bacon, and Isabela Le Bras

I think the revised manuscript is very good and am largely satisfied by the authors' responses to my comments. I have only a few remarks that I hope the authors will take into consideration before the manuscript is published.

I am still not fully convinced about the water mass transformation that appears to take place on the Greenland shelf in Figure 8a and the authors' response to this concern. The figure shows that densification by air-sea fluxes in the two lowest density bands (27.40-27.60 $\mathrm{kgm}^{-3}$ and 27.60-27.77 $\mathrm{kgm}^{-3}$) occurs on the Greenland shelf, which is typically still ice-covered in winter. The authors suggest that this could be summertime water mass transformation or that it is a consequence of remapping from temperature/salinity space to geographical space. I don't think that either of these suggestions can fully account for the water mass transformation shown in Figure 8. Firstly, in summer, I am unsure which process the authors refer to that would cause densification on the Greenland shelf. The air-sea interaction taking place would in general add buoyancy to the water column, particularly by solar insolation, not densify the water column. Mixing processes might, upwelling-favorable winds could for example bring dense water onto the shelf, but summertime air-sea interaction would in the mean reduce rather than increase density. Secondly, the Greenland shelf is primarily filled with Polar water masses. Within the domain considered in the manuscript, these water masses are found on the Greenland, Baffin, and Labrador shelves. It is not clear to me in which other regions densification by air-sea interaction in these temperature-salinity classes would occur, and then be remapped onto the Greenland shelf. I think it would be great if the authors could clarify this in the final version of the paper.

Line 27:
Labrador Sea Water should be capitalized.

Lines 499:
It should be "complementing" rather than "complimenting".

---

## Author Response (AR2)

**Dear Editor and Reviewers,**

Thank you for taking the time to once again review our manuscript. Please find our responses to each of your comments below. As previously, our responses are written in italic blue font. We have included a pdf showing the tracked changes in the following document:

**revised\_manuscript\_tracked\_changes.pdf**

This indicates where text has been added, deleted or replaced. The line numbers indicated below refer to the tracked changes document.

**Reviewer #1**

I am still not fully convinced about the water mass transformation that appears to take place on the Greenland shelf in Figure 8a and the authors' response to this concern. The figure shows that densification by air-sea fluxes in the two lowest density bands (27.40-27.60 kgm-3 and 27.60-27.77 kgm-3) occurs on the Greenland shelf, which is typically still ice-covered in winter. The authors suggest that this could be summertime water mass transformation or that it is a consequence of remapping from temperature/salinity space to geographical space. I don't think that either of these suggestions can fully account for the water mass transformation shown in Figure 8. Firstly, in summer, I am unsure which process the authors refer to that would cause densification on the Greenland shelf. The air-sea interaction taking place would in general add buoyancy to the water column, particularly by solar insolation, not densify the water column. Mixing processes might, upwellingfavorable winds could for example bring dense water onto the shelf, but summertime air-sea interaction would in the mean reduce rather than increase density. Secondly, the Greenland shelf is primarily filled with Polar water masses. Within the domain considered in the manuscript, these water masses are found on the Greenland, Baffin, and Labrador shelves. It is not clear to me in which other regions densification by air-sea interaction in these temperature-salinity classes would occur, and then be remapped onto the Greenland shelf. I think it would be great if the authors could clarify this in the final version of the paper.

We apologise for the confusion caused by our explanation of the surface forced water mass transformation over the Greenland Shelf. It's first worth re-iterating that the values shown in figure 8 are remapped from density space to geographical space. For the surface forced water mass transformation, we do this by assigning the transformation from a given density bin to each geographical point where the sea surface density falls within the range of the given density bin. This process is performed at each time-step, so that the fields shown in the manuscript represent the time-mean of the re-mapped fields. This implies that at each time-step the transformation for a given density bin is assigned equally to any location within density range of that bin, regardless of whether any surface fluxes acted at that location (see for example Figure 1 and 2 in this response for the EN4/ERA5 dataset combination). This is a caveat of the remapping process that is more detrimental for transformations in density space compared to temperature and salinity space.

An alternative approach for the surface forced water mass transformation is to average the water mass transformation before binning into density space (see for example Petit et al., 2020). In Figures 3 and 4, we show the seasonal average of this unbinned surface forced water mass transformation for the two lighter density ranges shown in Figure 8 of the manuscript using ERA5 and EN4. This highlights that while some water within these density ranges may exist where sea-ice is present, the surface buoyancy forcing in these regions of sea-ice cover is zero. That water within these density ranges exists where sea-ice is present, explains why the remapped transformations, given the caveat discussed above, incorrectly imply that surface fluxes act in these regions of sea-ice cover (Figure 1 and 2). Critically, as air-sea buoyancy fluxes do not act in these region of sea-ice cover, we therefore do not derive a surface forced water mass transformation where sea-ice is present. Further, our remapped transformations act only to aid visualisation of the transformation shown in tracer space.

We have chosen to continue to use the remapped the surface forced water mass transformations so that they remain consistent with the remapped residual water mass transformations also shown in the manuscript. To provide more clarity within the manuscript we have added some discussion on the caveat discussed above at lines 379-383 as follows:

"This also implies that the water mass transformation for a given tracer bin may be remapped to a region in which that transformation did not occur. For example, the water mass transformation by air--sea fluxes in a given tracer bin may be remapped into a region typically covered by sea ice where water within the range of the given tracer bin could exist. This caveat is more detrimental for the remapped diapycnal transformation due to the large isopycnal gradients of \$\Theta/S\$ in the subpolar North Atlantic and Nordic Seas."

Line 27: Labrador Sea Water should be capitalized.

**Corrected**

Lines 499: It should be "complementing" rather than "complimenting"

**Corrected**

**Editors corrections:**

L.176 "and we expect the we expect"

**Corrected**

L.565 "the in the"

Corrected